# A role for mitochondria–ER crosstalk in amyotrophic lateral sclerosis 8 pathogenesis

Cathal Wilson[1,2,*], Laura Giaquinto[1,2,*], Michele Santoro[1], Giuseppe Di Tullio[1], Valentina Morra[1], Wanda Kukulski[3,4], Rossella Venditti[1,2], Francesca Navone[5], Nica Borgese[5], Maria Antonietta De Matteis[1,2]

**Protein aggregates in motoneurons, a pathological hallmark of amyotrophic lateral sclerosis, have been suggested to play a key pathogenetic role. ALS8, characterized by ER-associated inclusions, is caused by a heterozygous mutation in VAPB, which acts at multiple membrane contact sites between the ER and almost all other organelles. The link between protein aggregation and cellular dysfunction is unclear. A yeast model, expressing human mutant and WT-VAPB under the control of the orthologous yeast promoter in haploid and diploid cells, was developed to mimic the disease situation. Inclusion formation was found to be a developmentally regulated process linked to mitochondrial damage that could be attenuated by reducing ER–mitochondrial contacts. The co-expression of the WT protein retarded P56S-VAPB inclusion formation. Importantly, we validated these results in mammalian motoneuron cells. Our findings indicate that (age-related) damage to mitochondria influences the propensity of the mutant VAPB to form aggregates via ER–mitochondrial contacts, initiating a series of events leading to disease progression.**

## Introduction

Amyotrophic lateral sclerosis (ALS) is a heterogeneous group of fatal, progressive, neurodegenerative disorders that typically start in middle adulthood and are characterized by the death of motoneurons. Several causative genes are associated with ALS, the most common being *SOD1*, *FUS*, *TARDBP*, and *C9ORF72*, although many putative causal genes have also been described (Mejzini et al, 2019). The large number of genes and cellular processes implicated in ALS implies that multiple factors could contribute to disease progression whose pathogenic mechanisms remain unclear.

ALS type-8 (ALS8) is a rare form of ALS caused by mutations in the *VAPB* gene (Nishimura et al, 2004). VAPB is a C-tail–anchored membrane protein of the ER comprising a cytosolic N-terminal domain, homologous to the nematode major sperm protein (MSP), a central coiled-coil domain, and a C-terminal transmembrane domain (Murphy & Levine, 2016). Through its N-terminal MSP domain, VAPB interacts with a large number of proteins (Murphy & Levine, 2016) and thus has a role in multiple cellular functions such as interorganellar lipid exchange, Ca²⁺ homeostasis, membrane traffic, cytoskeleton organization, autophagy, mitochondrial function, and neurite outgrowth (Kamemura & Chihara, 2019; Borgese et al, 2021a). Many of these interactions and functions occur at membrane contact sites (MCSs) where VAPB acts as a component of tethering complexes between different organelles (De Vos et al, 2012; Dong et al, 2016; Venditti et al, 2019), a function that has been conserved by VAPB orthologs throughout evolution from yeast to mammals (Manford et al, 2012; Murphy & Levine, 2016). Extensive communication between organelles is increasingly appreciated as a crucial player in cell physiology, with relevance to metabolic diseases, aging, and neurodegeneration (Tubbs & Rieusset, 2017; Petkovic et al, 2021).

A single point mutation from proline to serine at position 56 of VAPB (P56S-VAPB) causes dominantly inherited ALS8 (Nishimura et al, 2004). This single point mutation, which is located in the cytosolic MSP domain, is considered to lead to protein misfolding with the generation of inclusions, which correspond to altered ER structures (Teuling et al, 2007; Fasana et al, 2010; Papiani et al, 2012). These may result from the generation of tightly apposed ER cisternae through interactions between the mutant cytosolic domains (Fasana et al, 2010; Papiani et al, 2012). The consequences of the P56S mutation have been reported to be alterations in autophagy (Larroquette et al, 2015; Zhao et al, 2018), unfolded protein response (UPR) (Kanekura et al, 2006), and lysosomal (Mao et al, 2019) and mitochondrial (Mórotz et al, 2012) physiology, reflecting the many VAPB interactors and sites of action in the cell. The importance of each of these defective processes in the pathogenesis of ALS8, or whether they all contribute, is still a matter of debate. Similarly, the mechanisms underlying the dominant inheritance of P56S-VAPB are unclear. The P56S-VAPB inclusions have been reported to sequester the WT protein leading to loss of function resulting from a

---

[1]Telethon Institute of Genetics and Medicine, TIGEM, Pozzuoli, Italy   [2]Department of Molecular Medicine and Medical Biotechnology, University of Naples Federico II, Naples, Italy   [3]Institute of Biochemistry and Molecular Medicine, University of Bern, Bern, Switzerland   [4]MRC Laboratory of Molecular Biology, Cambridge, UK   [5]CNR Neuroscience Institute, Vedano al Lambro, Italy

Correspondence: dematteis@tigem.it; cathalwilson636@gmail.com
*Cathal Wilson and Laura Giaquinto contributed equally to this work

dominant negative effect (Teuling et al, 2007; Ratnaparkhi et al, 2008; Suzuki et al, 2009) or by the sequestration of functionally important VAPB interactors (Forrest et al, 2013; Kuijpers et al, 2013). However, many investigations on the pathogenic mechanisms of P56S-VAPB have been analyzed in cultured cells acutely over-expressing the WT and the mutant protein (reviewed in Navone et al [2015]). This could produce effects that do not necessarily reflect the situation in cells where the mutant protein is chronically expressed from a single allele, leading to constitutive inclusion formation or pathway deregulation that is exacerbated by overexpression.

An alternative scenario is that P56S-VAPB's instability leads to haploinsufficiency. In support of this possibility, partial depletion of WT-VAPB, even in the absence of the mutated protein, alters phosphatidylinositol-4-phosphate levels in the Golgi apparatus and in a population of acidic vesicles and reduces neurite extension of NSC-34 motoneuron-like cells when these are induced to differentiate (Genevini et al, 2019). VAPB protein levels were also reported to be reduced in ALS8 patient IPSC-derived motoneurons (Mitne-Neto et al, 2011) and in cellular and transgenic animal models (reviewed in Borgese et al [2021b]). Thus, neurons may require the full complement of VAPB for proper function.

To gain insight into the mechanism of generation of P56S-VAPB inclusions and their pathogenic role, we developed a novel ALS8 yeast model: we engineered *Saccharomyces cerevisiae* to express human WT-VAPB or P56S-VAPB under the control of the promoter and terminator of the yeast homolog *SCS2*, at the *SCS2* locus, in order to prevent the overexpression of the transfected proteins. Yeast models have been useful for gaining insights into neuro-degenerative diseases, including different forms of ALS, which have been subsequently validated in mammalian systems (Oliveira et al, 2017; Ruetenik & Barrientos, 2018). Our results, supported by studies in mammalian motoneuron cells, suggest that inclusion formation is due to mitochondrial dysfunction whose consequences are relayed via ER–mitochondrial MCSs and that the WT protein, rather than being sequestered by the inclusions, slows down their formation. These findings are discussed in the context of the progressive late-onset neuronal etiology of ALS8.

# Results

### A yeast model for ALS8

A synthetic DNA fragment was generated consisting of the human *VAPB* coding sequence (codon-optimized for expression in yeast) with monomeric GFP inserted between the coiled-coil and the transmembrane domains under the control of the yeast *SCS2* promoter and terminator (Fig 1A and Table S1A). Both WT and mutant (P56S) versions of this construct were inserted into the yeast genome via homologous recombination to replace the endogenous *SCS2* gene. The WT-VAPB-GFP, but not P56S-VAPB-GFP, could partially complement the inositol auxotrophy of *scs2Δ* cells (Kagiwada et al, 1998) (Fig S1A), in keeping with the previously described functionality of the WT and mutant mammalian proteins expressed in yeast (Suzuki et al, 2009; Stump et al, 2023).

When expressed in yeast, human VAPB-mGFP showed the same localization pattern as the endogenous Scs2 protein (Chao et al, 2014; Neller et al, 2015; Hoffmann et al, 2019), being found in both the cortical and perinuclear ER, at the bud tip and the septum, and frequently showing more intense fluorescence at the nuclear–vacuolar junction (Fig 1B). The mutant protein exhibited the same localization pattern as the WT protein, and no indications of protein aggregation were observed during log-phase growth, that is, a time of exponential growth when nutrients are plentiful (Fig S1B). However, we found that P56S-VAPB-mGFP formed bright foci in some cells upon entry into the stationary phase, a stage where nutrients are exhausted and cells stop dividing, a phenomenon that was never observed for the WT protein (Fig 1C). Hereon, we refer to these bright foci as inclusions in keeping with the nomenclature in mammalian cells and our EM analysis (Fasana et al, 2010, and see below).

### Inclusions form in the non-quiescent population of stationary-phase cells

Yeast cells that exhaust their nutrient supply go through the diauxic shift and enter into the stationary phase, which is characterized by two major cell populations, namely, quiescent (Q) and non-quiescent (NQ). Quiescent cells are mostly unbudded daughter cells that retain both viability and the ability to reproduce, and synchronously reenter the mitotic cell cycle suggesting that they are in a G0 state. The non-quiescent population is heterogeneous, contains both budded and unbudded cells, does not synchronously reenter mitosis, and loses the ability to reproduce over time (Allen et al, 2006).

We noted that the stationary-phase cells with P56S-VAPB-mGFP inclusions corresponded to cells with large vacuoles (Fig 1C). Because the NQ but not the Q population has large vacuoles (Allen et al, 2006; Lee et al, 2016), we considered the possibility that the former might correspond to those with inclusions. We therefore separated Q and NQ cells on a Percoll gradient and found that NQ cells did indeed contain inclusions, while the Q cells did not (Fig 1D). Entry into the stationary phase is accompanied by major metabolic changes (De Virgilio, 2012), and one of the major features that distinguishes Q from NQ cells is the state of the mitochondria. Q cells, while not dividing, have numerous small vesicular and active mitochondria with high rates of respiration. NQ cells, in contrast, have few large globular mitochondria and appear to be respiration-deficient with the down-regulation of some proteins involved in oxidative phosphorylation and an increase in ROS (Davidson et al, 2011; Laporte et al, 2018). To monitor changes in mitochondrial morphology, we inserted a mitochondrial reporter, mitochondrial targeting sequence (MTS)-Cherry, into the genome of VAPB-expressing cells. No obvious alterations in mitochondrial morphology were observed in log-phase P56S-VAPB-mGFP–expressing cells (Fig S1B). Instead, in the stationary phase, the cells with inclusions had globular mitochondria, whereas those without inclusions had vesiculated mitochondria (Fig 1D), corresponding to the description of the mitochondria in NQ and Q cells, respectively (Davidson et al, 2011; Laporte et al, 2018). This supports the conclusion that only non-quiescent cells form inclusions.

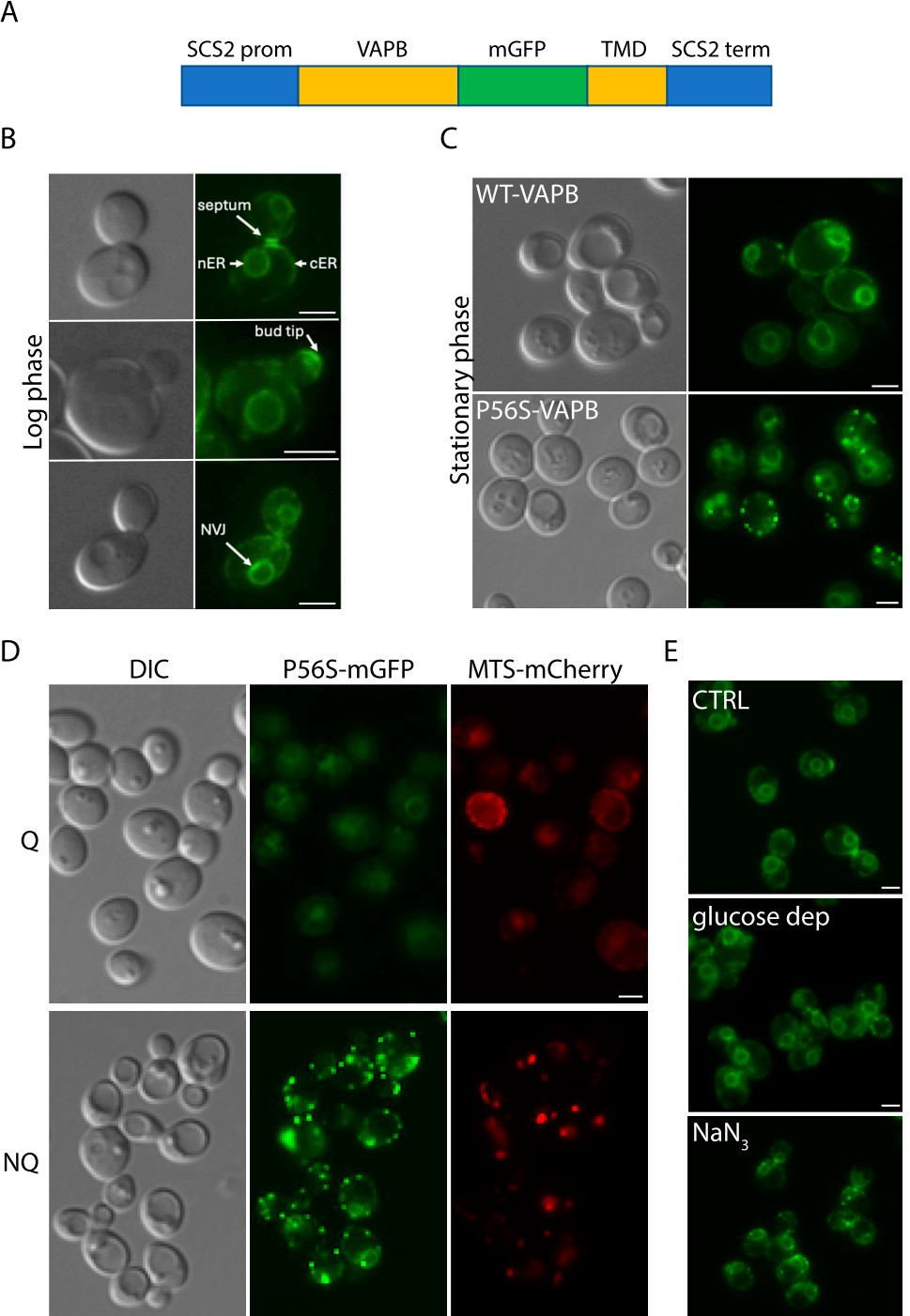

**Figure 1. Yeast model for ALS8.**
**(A)** Schematic representation of the synthetic VAPB construct containing the human *VAPB* coding sequence, with monomeric GFP inserted before the transmembrane domain (TMD), under the control of the *SCS2* promoter and terminator. **(B)** Localization of WT-VAPB-mGFP after insertion into the genome to replace the yeast *SCS2* gene. Sites of major fluorescence intensity are indicated. nER, perinuclear ER; cER, cortical ER; NVJ, nuclear–vacuole junction. Scale bars, 2 μm. **(C)** Stationary-phase cells expressing WT-VAPB-mGFP or P56S-VAPB-mGFP. Scale bars, 2 μm. **(D)** VAPB inclusions form in the NQ population of stationary-phase cells. Cells expressing P56S-VAPB-mGFP and MTS-Cherry (to visualize mitochondria) were grown to the stationary phase and then subjected to a Percoll gradient to separate quiescent (Q) from non-quiescent (NQ) cells. Scale bars, 2 μm. **(E)** P56S-VAPB-mGFP–expressing log-phase cells left untreated (CTRL) or subjected to acute glucose depletion (glucose dep) or treatment with NaN$_3$ (0.5 mM, 60 min). Scale bars, 2 μm.

Strikingly, calorie-restricted cells in the stationary phase retained the mitochondrial morphology of quiescent cells while suppressing the formation of inclusions (Fig S1C), again correlating mitochondrial function, whose biogenesis and bioenergetic efficiency are improved by calorie restriction (López-Lluch et al, 2006), with inclusion formation.

We exploited the lack of inclusions during the log phase to look for conditions that might induce their formation. An obvious candidate was a lack of glucose because this is consumed completely as cells go through the diauxic shift and enter the stationary phase. Indeed, we found that in many—but not all—cells, acute glucose depletion led to the formation of inclusions, which appeared to be associated with the perinuclear and cortical ER

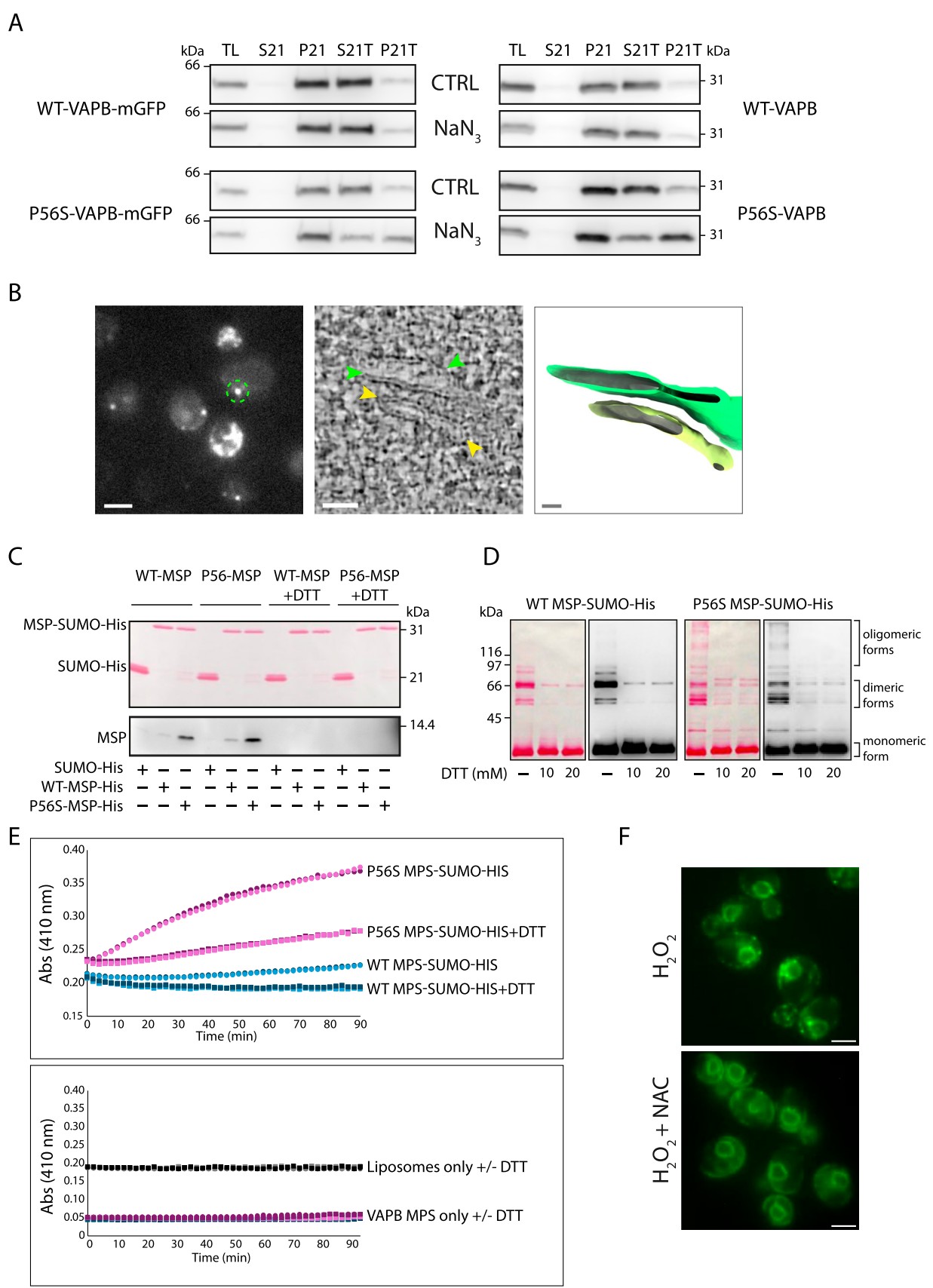

(Fig 1E). Because there seemed to be a link to dysfunctional mitochondria, we tested the effects of the electron-transport inhibitor sodium azide (NaN₃) and found that it strongly induced inclusion formation, even in the presence of glucose (Fig 1E). We note that WT-VAPB-mGFP never formed inclusions under any of the conditions that we have tested in this study. Before proceeding with the analysis of this response, we first asked whether these inclusions have the characteristics of mammalian inclusions.

## Yeast VAPB inclusions have characteristics of mammalian inclusions

We tested whether the inclusions observed in yeast had characteristics similar to those described in mammalian cells, such as resistance to Triton X-100 extraction (Suzuki et al, 2009; Moumen et al, 2011). Yeast cells expressing untagged or mGFP-tagged WT-VAPB or P56S-VAPB were left untreated or treated with NaN₃ and then analyzed for Triton X-100 resistance (see the Materials and Methods section). The P56S protein but not the WT showed resistance to Triton X-100 extraction only after treatment with NaN₃ (Fig 2A).

In mammalian cells, P56S-VAPB inclusions have been shown to correspond to paired ER cisternae (Fasana et al, 2010). We used correlative light and electron microscopy (CLEM) on the P56S-VAPB-mGFP–expressing yeast cells after induction with NaN₃ and found that the inclusions corresponded to two paired ER cisternae (Fig 2B). These structures qualitatively recall those induced by P56S in mammalian cells, although the paired cisternae showed a less extensive contact area when compared to the OSER described by Fasana et al (2010), possibly because of the lower levels of the expression of P56S-VAPB in our yeast model.

The pairing of ER cisternae may occur in trans by misfolding of the MSP domain because of the P56S mutation leading to the exposure of hydrophobic patches that facilitates aberrant oligomerization (Kim et al, 2010). We tested this possibility using multiple approaches. First, we evaluated the ability of the MSP domains to oligomerize using in vitro pull-down assays. SUMO-His-tagged WT or P56S MSP domains were attached to Ni beads and used as baits to pull down untagged MSP domains. As shown in Fig 2C, the P56S MSP domain was more efficiently captured by the immobilized fusion proteins than its WT counterpart, and this association could be counteracted by reducing agents (Fig 2C).

Second, SUMO-His-tagged MSP domains from WT and P56S-VAPB were purified from *E. coli* Rosetta (DE3) cells and analyzed on polyacrylamide gels in the presence or absence of DTT. Both Ponceau S staining and Western blot analysis showed that the WT MSP was mostly in its monomeric (30.2 kD) and dimeric (60 kD) forms, whereas the P56S MSP appeared to associate in multiple forms that ran at high molecular weights (Fig 2D). Reducing agents, such as DTT, led to a reduction of the multimeric forms of P56S-VAPB MSP.

Finally, we assessed the propensity of WT or mutant MSP domain–covered liposomes to aggregate. The P56S mutation but not the WT protein led to liposome aggregation, which was reverted by reducing agents (Fig 2E). Our results support the hypothesis that trans pairing of misfolded VAPB MSP domains, favored by an oxidative environment, contributes to ER pairing, which could correspond to the inclusions observed at the EM level and by fluorescence microscopy in NQ cells and in cells treated with NaN₃. In support of this hypothesis, we found that treatment of P56S-VAPB-mGFP–expressing yeast cells with $H_2O_2$ induced inclusion formation, which could be rescued by the antioxidant N-acetylcysteine (Fig 2F).

## WT-VAPB retards inclusion formation

Another characteristic of the inclusions in mammalian cells is that the P56S-VAPB aggregates can sequester the WT protein, which is found both in Triton-insoluble P56S-VAPB aggregates and in inclusions observed by immunofluorescence (Teuling et al, 2007; Ratnaparkhi et al, 2008; Suzuki et al, 2009). The observation that VAPB molecules can oligomerize (Kim et al, 2010) may mean that the in trans pairing of ER cisterna by P56S-VAPB traps the WT molecule in these inclusions. However, it has been suggested that in the systems where it has been reported, the sequestration of the WT molecule may be due to protein overexpression (Navone et al, 2015).

We therefore exploited the yeast model using diploid cells in which WT and mutant proteins are co-expressed from single alleles, always under the control of the endogenous promoter of the yeast ortholog, thus mimicking the heterozygous situation of ALS8. WT and P56S versions of VAPB were tagged with a RUBY fluorescent marker replacing the mGFP, inserted into the genome, and then mated with the mGFP-tagged strains to produce a heterozygous

**Figure 2. Yeast inclusions exhibit characteristics of mammalian inclusions.**
**(A)** Cells expressing either GFP-tagged (left panels) or untagged (right panels) WT-VAPB or P56S-VAPB were left untreated (CTRL) or treated with NaN₃ (0.5 mM, 60 min) and then processed for Triton X-100 resistance. Samples were processed for Western blot analysis and detected with an anti-VAPB antibody. TL = total lysate; S21 = 21,000*g* supernatant; P21 = 21,000*g* pellet (microsomal fraction); S21T = 21,000*g* supernatant after Triton X-100 extraction of the P21 pellet; P21T = 21,000*g* pellet after Triton X-100 extraction of the P21 pellet. **(B)** CLEM showing paired ER cisternae. Left, fluorescence image of a resin section of cells expressing P56S-VAPB-mGFP treated with NaN₃. The circled inclusion was targeted by CLEM. Middle, virtual slice through electron tomogram corresponding to the circled structure. Yellow arrows and green arrows indicate two stacked structures. Right, a segmentation model of the stack colored according to the arrows in the middle panel. Scale bars from left to right: 2 μm, 50 nm, 25 nm. **(C)** In vitro pull-down assays. SUMO-His-tagged WT or P56S MSP domains, or SUMO-His alone, attached to Ni beads, were incubated with untagged WT or P56S MSP domains, with or without DTT. After washing, the beads were treated with gel loading buffer, separated by PAGE, and transferred to nitrocellulose. Top panel, Ponceau S staining of the SUMO-His proteins. Bottom panel, the area of the membrane corresponding to the untagged MSP domains at ~14 kD was immunodetected with an anti-VAPB antibody. **(D)** Ponceau S staining and Western blot analysis of SUMO-His-tagged WT or P56S-VAPB MSP domains in the presence of DTT (10 and 20 mM). The P56S MSP domain associates in multiple oligomeric forms, which are not present for the WT MSP domain. **(E)** Liposome aggregation assays. Top panel, WT or mutant MSP domain–covered liposomes were incubated with or without DTT. The absorbance (OD) at 410 nm was measured every 2 min over a period of 90 min using Neo2 Microplate Reader. Bottom panel, the same assay with liposomes or the VAPB MSP domains incubated separately. **(F)** P56S-VAPB-mGFP yeast cells were treated with $H_2O_2$ in the presence or absence of the antioxidant N-acetylcysteine (NAC). Scale bars, 2 μm.
Source data are available for this figure.

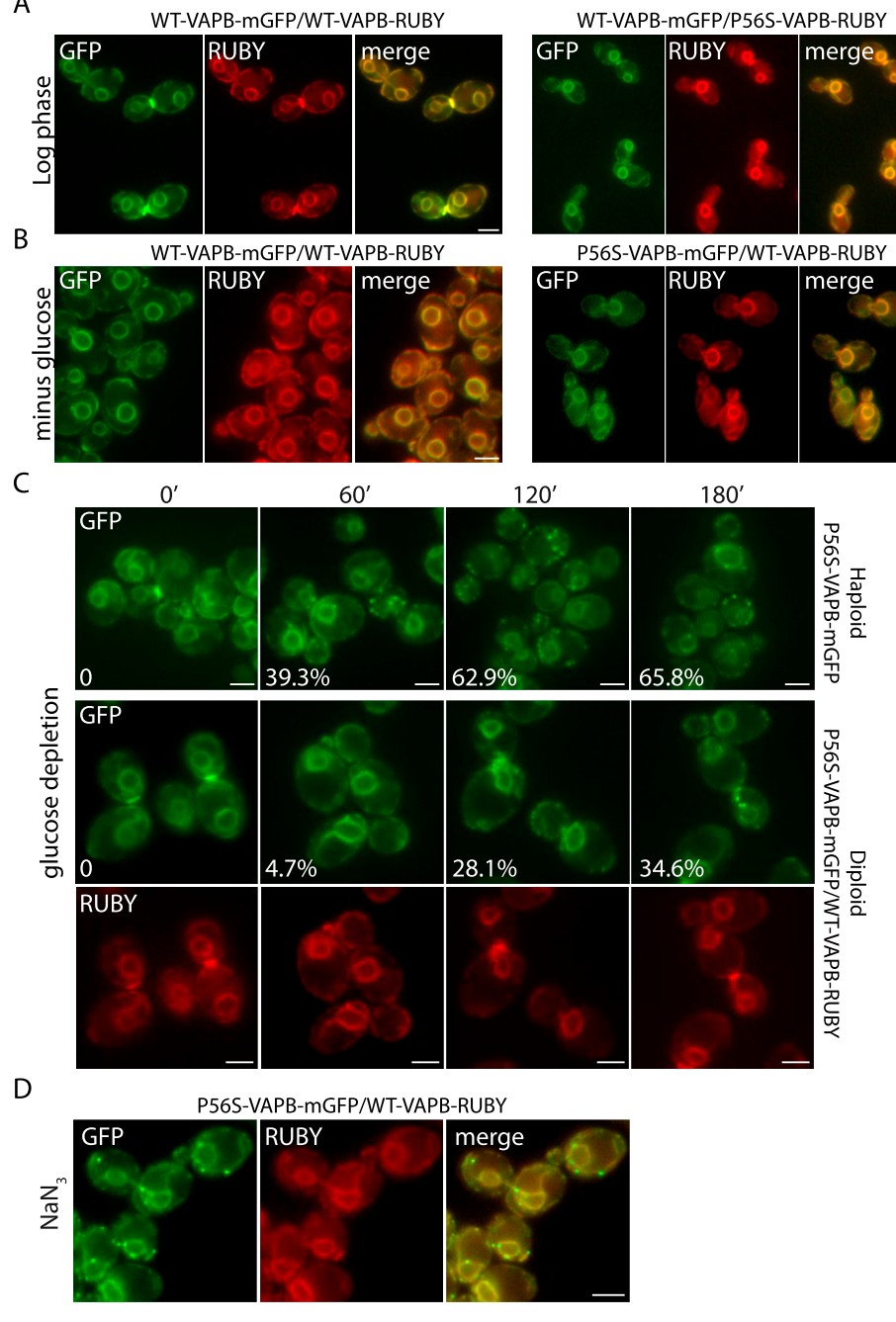

**Figure 3. WT-VAPB slows down inclusion formation in yeast cells.**
**(A)** Fluorescence images of log-phase diploid yeast cells expressing differentially tagged VAPB (WT-VAPB-GFP with WT-VAPB-RUBY and WT-VAPB-GFP with P56S-VAPB-RUBY). Scale bars, 2 μm. **(B)** Diploid cells expressing differentially tagged VAPB (WT-VAPB-GFP with WT-VAPB-RUBY or P56S-VAPB-GFP with WT-VAPB-RUBY) were glucose-depleted for 60 min. Scale bars, 2 μm. **(C)** Haploid cells expressing P56S-VAPB-GFP or diploid cells expressing P56S-VAPB-GFP and WT-VAPB-RUBY were glucose-depleted, and inclusion formation was monitored over time (min). Inclusions form in haploid cells after 60 min but take up to 120 min to form in the diploid. Numbers in the panels indicate the percentage of cells forming inclusions at each time point. Scale bars, 2 μm. **(D)** Diploid cells expressing P56S-VAPB-GFP and WT-VAPB-RUBY were treated with 0.5 mM NaN₃, 120 min. Scale bars, 2 μm.

diploid with a P56S-VAPB-mGFP/WT-VAPB-RUBY combination, as well as homozygous diploids expressing differentially tagged WT and mutant versions (see the Materials and Methods section for details). Both fluorescent markers showed a perfect overlap (Fig 3A). In the log phase, no inclusions were observed in the heterozygous diploids or in the WT homozygote, as expected (Fig 3A). After 60 min of glucose depletion, inclusions were observed in the homozygous diploid that expressed two copies of P56S-VAPB (Fig S2), but not in the heterozygote or in the homozygote expressing two copies of WT-VAPB (Figs 3B and S2), suggesting that the WT copy is actually repressing inclusion formation. Indeed, compared with the heterozygous diploids, where inclusions appear after 120 min of glucose depletion, inclusions formed faster and were readily observable in the haploid strain after 60 min (Fig 3C). The WT protein did not appear to be recruited into the inclusions in the heterozygous diploid even after a longer time of glucose depletion (Fig 3C) or after induction with 0.5 mM NaN₃ for 60 (Fig S2) or 120 min (Fig 3D). However, at this resolution, we cannot exclude that some WT protein is sequestered, and indeed, at longer times the formation of stable P56S-insoluble aggregates might be expected to retain the WT protein.

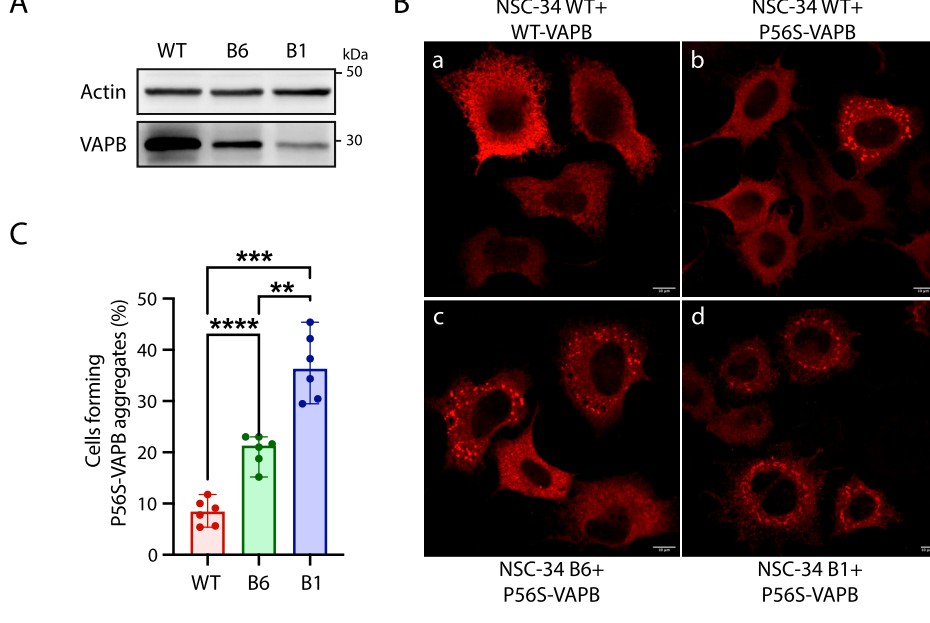

**A**

WT | B6 | B1 | kDa
Actin ———— — 50
VAPB ———— — 30

**C**

Cells forming P56S-VAPB aggregates (%)

(bar graph with WT, B6, B1; significance ****, **, ***)

**B**

NSC-34 WT+ WT-VAPB (a) | NSC-34 WT+ P56S-VAPB (b)

NSC-34 B6+ P56S-VAPB (c) | NSC-34 B1+ P56S-VAPB (d)

**D**

NSC-34 B1 cells forming P56S-VAPB aggregtes (%)

(bar graph CTRL vs VAPA KD, **)

Figure 4. WT-VAPB slows down inclusion formation in mammalian cells. **(A)** Western blot analysis of protein extracts from WT, B6, and B1 NSC-34 cell lines (B6 and B1 were previously determined to express 40% and 10% of residual VAPB expression, respectively; Genevini et al, 2019). The blot was detected with an anti-VAPB antibody, and an anti-actin antibody was used as a loading control. **(B)** Representative images of WT NSC-34 cells transfected with (a) WT-VAPB or (b) P56S-VAPB, and (c) B6 and (d) B1 NSC-34 cells transfected with P56S-VAPB. The cells were immunostained with an anti-VAPB antibody. Scale bars, 10 $\mu$m. **(B, C)** Quantification of the percentage of P56S-VAPB–transfected cells in (B) showing P56S-VAPB aggregates. N = 6, n > 100. Statistical significance among groups was calculated by Welch's one-way ANOVA with Dunnett's post hoc test for multiple comparisons. **P < 0.01, ***P < 0.001, ****P < 0.0001. **(D)** Quantification of the percentage of P56S-VAPB–transfected B1 cells forming aggregates in control (CTRL) or under VAPA down-regulation (VAPA KD). N = 3, n > 100. **P < 0.01, unpaired t test. Source data are available for this figure.

To test the effect of WT-VAPB on P56S inclusion formation in mammalian cells, we resorted to two stable motoneuron-like NSC-34 VAPB-silenced lines that have ~40% and 10% residual VAPB expression (Genevini et al, 2019). The WT NSC-34 cell line (a motoneuron-like cell line) and the two NSC-34 VAPB-silenced lines (B6 and B1, Fig 4A) were transfected with P56S-VAPB (Fig 4B), and the level of inclusion formation was quantified. The percentage of non-silenced NSC-34 cells that spontaneously form P56S-VAPB inclusions was only about 10% (Fig 4C), much less than when P56S-VAPB is expressed in HeLa cells, making them more amenable to investigation. We found that the level of inclusion formation was inversely correlated with endogenous VAPB levels (Fig 4C), corroborating the findings in yeast that the WT protein could act to suppress aggregate formation. Given the ability of VAPB to hetero-oligomerize with VAPA, we also tested the possibility that VAPA could play a role in controlling the rate/extent of P56S-VAPB inclusion formation. Thus, we assessed the impact of decreasing the levels of VAPA (by siRNA) on the P56S-VAPB inclusions in the B1 clone (expressing low levels of WT-VAPB). We found that inclusion formation occurred to a higher extent upon VAPA down-regulation (Fig 4D), suggesting that heterodimerization of P56S-VAPB with VAPA might partially counteract the formation of P56S-VAPB aggregates.

Considering the data from both yeast and mammalian cells, we conclude that an important and hitherto unreported effect of the interactions between the WT and the mutant protein might be to retard inclusion formation. This aspect is addressed further in the discussion.

### Inclusions can be induced by inhibiting mitochondrial function

Having a system that could induce P56S aggregates with characteristics that reflect those observed in mammalian cells and given the indication that mitochondrial dysfunction might be a contributing factor, we decided to investigate the interplay between mitochondrial function and inclusion formation in more detail.

As described above, glucose depletion can induce aggregate formation. This was not due solely to an acute depletion of the carbon source, however, because an acute depletion of galactose did not have the same effect (Fig 5A–C). ATP levels drop rapidly to a low level in cells abruptly depleted of glucose, whereas a more modest decrease occurs when there is an acute depletion of galactose (Ashe et al, 2000; Xu & Bretscher, 2014). Glucose-grown cells principally use fermentation for growth, whereas galactose-grown cells use both fermentation and respiration with increased mitochondrial biogenesis (Wallace et al, 1968; Nagata et al, 1975; Fendt &

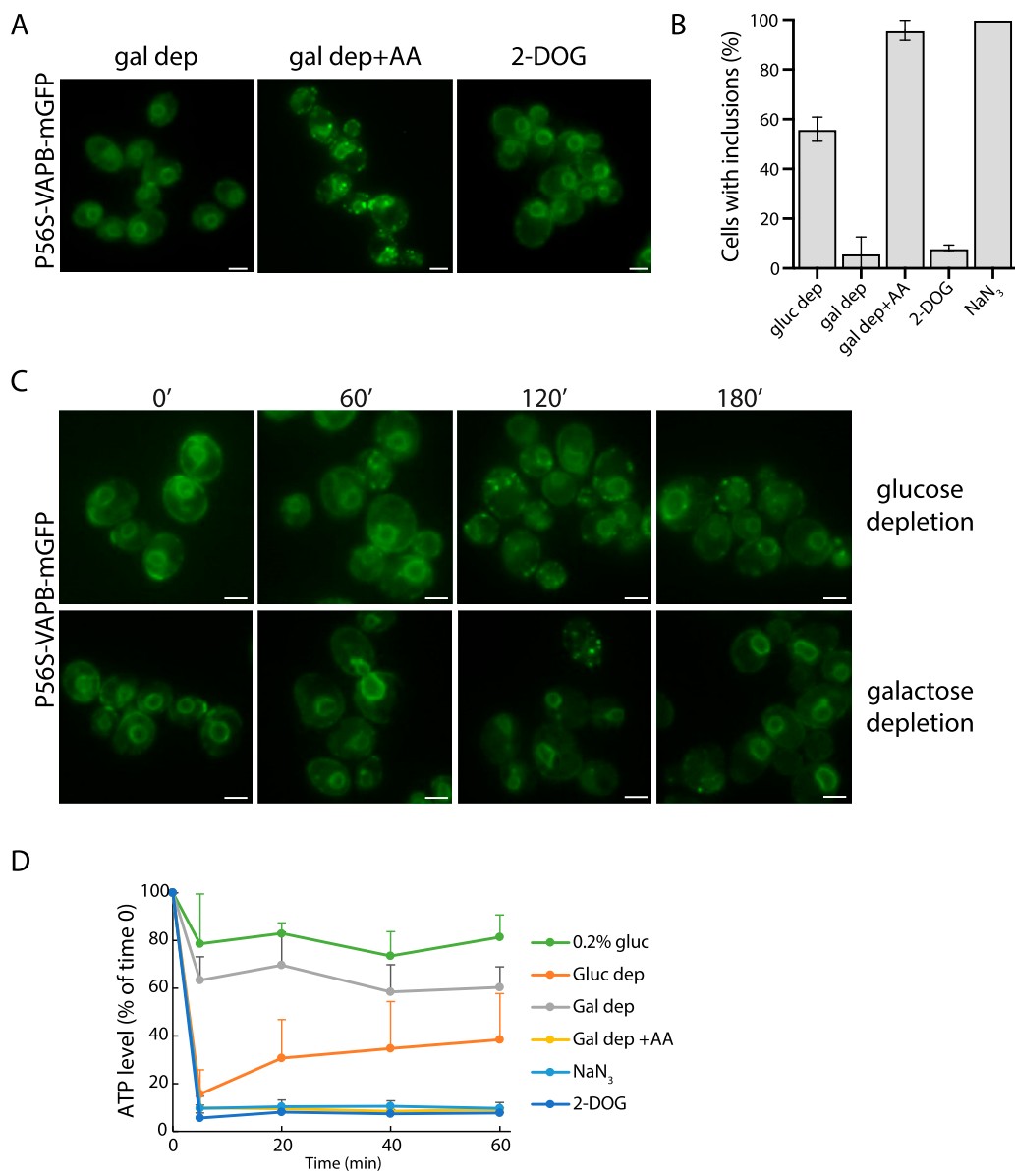

**Figure 5.  Mitochondrial damage, but not a drop in ATP levels, induces P56S-VAPB inclusion formation.**
**(A)** P56S-VAPB-GFP cells were subjected to acute galactose depletion (gal dep) and acute galactose depletion with the addition of antimycin A (gal dep + AA) or grown in 0.2% glucose and treated with 2-deoxyglucose (2-DOG; see the Materials and Methods section). Scale bars, 2 $\mu$m. **(A, B)** Quantification of the percentage of cells with inclusions under the conditions described in (A) and in glucose-depleted cells (gluc dep) and NaN$_3$-treated cells (0.5 mM, 60 min), as shown in Fig 1E. Mean ± SD. N = 3, n > 150. **(C)** Time course of inclusion formation after acute glucose or galactose depletion. Cells expressing P56S-VAPB-mGFP were grown to the log phase in glucose or galactose as a carbon source, washed, and resuspended in SC (YNB + amino acids, minus a carbon source) for the indicated times (min). Scale bars, 2 $\mu$m. **(D)** ATP levels were measured at the indicated times under the indicated conditions and expressed as a percentage of the ATP level at T = 0 (set at 100%). ATP levels are shown for SD containing 0.2% glucose as a control for the 2-DOG treatment (see the Materials and Methods section). N = 3; data are presented as the mean ± SEM.
Source data are available for this figure.

Sauer, 2010). Thus, considering the possibility of an involvement of mitochondria in the induction of inclusions, we treated galactose-depleted cells with antimycin A, an inhibitor of the respiratory chain, and this led to a strong induction of inclusion formation (Fig 5A and B), in keeping with the observation above that the electron-transport inhibitor (NaN$_3$) is also a strong inducer of inclusions.

These data supported an involvement of mitochondrial dysfunction in the formation of the P56S-VAPB inclusions, which suggested the possibility that a severe drop in ATP levels might be responsible for the induction of inclusions. In accordance with this, we observed a rapid drop in ATP levels upon glucose depletion, upon antimycin A treatment of galactose-depleted cells, or after NaN$_3$ treatment, whereas galactose depletion alone led to only an ~50% reduction (Fig 5D). However, ATP levels started to recover in the glucose-depleted cells during the time course of the experiment (60 min), in accordance with previous reports (Ashe et al,

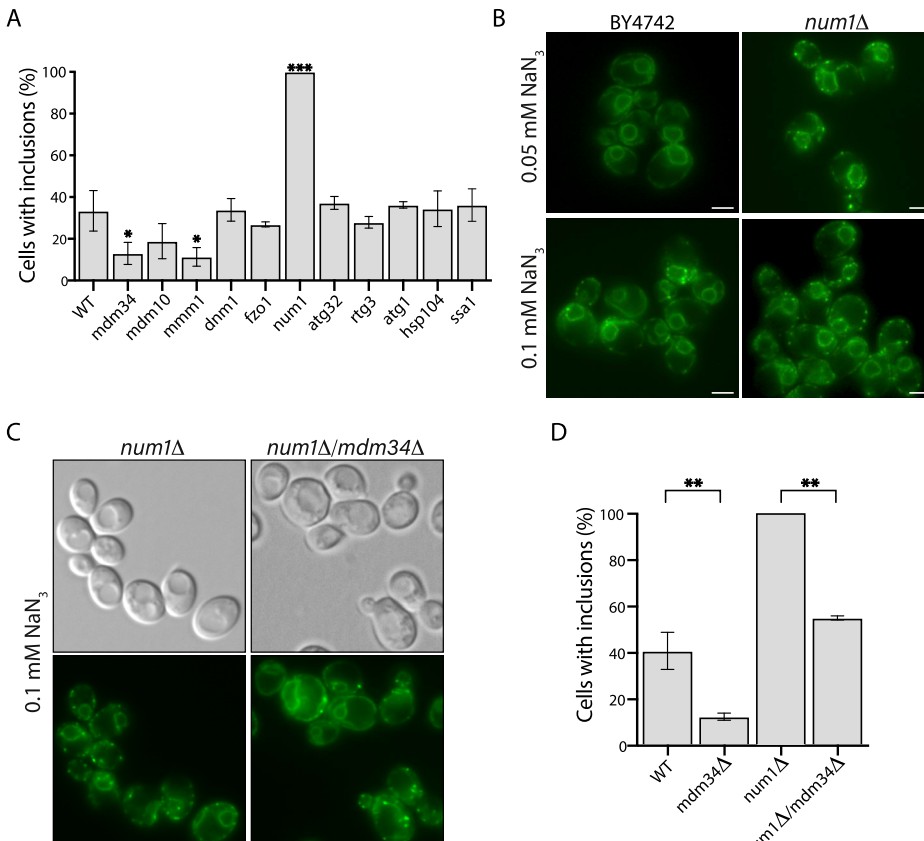

**Figure 6. Disruption of ER–mitochondrial contact sites in yeast attenuates inclusion formation.**
**(A)** P56S-VAPB-GFP construct was inserted into the genome of the indicated gene deletion strains and analyzed for inclusion formation using 0.1 mM NaN₃. The percentage of cells with inclusions is shown, mean ± SD. N = 3, n > 300. *P < 0.1, ***P < 0.005, relative to WT, unpaired t test.
**(B)** Hypersensitivity to inclusion induction in num1Δ cells. P56S-VAPB-GFP–expressing cells with an otherwise WT genetic background (BY4742) or with the NUM1 gene deleted (num1Δ) were treated with 0.05 mM or 0.1 mM NaN₃. Scale bars, 2 μm. **(C)** ERMES deletion reduces the hypersensitivity of num1Δ cells. Cells expressing P56S-VAPB-GFP with deletion of NUM1 (num1Δ) or NUM1 and the ERMES subunit gene MDM34 (num1Δ/mdm34Δ) were treated with 0.1 mM NaN₃. Scale bars, 2 μm. **(D)** Quantification of the percentage of cells with P56S-VAPB inclusions in the indicated strains, mean ± SD. N = 3, n > 300. **P < 0.05, unpaired t test.
Source data are available for this figure.

2000; Takaine et al, 2019), and thus did not correlate with the timeframe of inclusion formation (Fig 5C). We then tested 2-deoxyglucose, which causes glucose repression but cannot be further broken down by glycolysis to yield ATP. 2-Deoxyglucose treatment led to a rapid and sustained drop in ATP, as reported previously (Takaine et al, 2019), but not to significant inclusion formation (Fig 5A–C). Thus, we could not find a correlation between ATP levels and the level of inclusion formation.

We noted that the inducing conditions in the log phase led to cells with enlarged vacuoles, reminiscent of the large vacuoles in NQ cells. This could lead to molecular crowding, but we found no correlation between molecular crowding (Fig S3A) and vacuolar pH (Fig S3B and C) with inclusion formation. Because VAPB is an ER protein, we also considered its role in the UPR but found that inclusion formation is not induced by, nor does it induce, the UPR (Fig S4A–C).

A further consequence of the conditions that induce inclusions is mitochondrial fragmentation (Fig S4D). Given the close association between the ER and mitochondria and that in mammalian cells, VAPB interacts directly with mitochondria via PTPIP51 (De Vos et al, 2012), mitochondrial fragmentation might have affected the folding capacity of the mutated protein. This was excluded, however, because low concentrations of NaN₃ could fragment the mitochondria without the formation of inclusions (Fig S4D), whereas NaN₃ treatment of P56S-VAPB-mGFP/dnm1Δ cells, where the dynamin-related GTPase DNM1 was deleted and as a result is resistant to

NaN₃-induced mitochondrial fragmentation (Fekkes et al, 2000), still induces inclusion formation (Fig S4E).

Glucose depletion did not dissipate the mitochondrial membrane potential, whereas NaN₃ treatment reduced but did not eliminate it (Fig S4F), indicating that a lack of mitochondrial membrane potential was not associated with inclusion formation.

### Disruption of ER–mitochondrial MCS affects inclusion formation

The correlation between inclusion formation and mitochondrial function in non-quiescent cells and the strong induction of inclusions by mitochondrial inhibitors pointed to an involvement of mitochondria. To dissect the molecular mechanisms underlying this involvement, we decided to use a genetic approach, generating a series of P56S-VAPB-mGFP–expressing gene deletion mutants that impact on mitochondrial function such as fusion (fzo1Δ), fission (dnm1Δ), mitophagy (atg32Δ), ER–mitochondrial contacts (mdm34Δ, mdm10Δ, mmm1Δ), PM–mitochondrial contact sites (num1Δ), and retrograde signaling (rtg3Δ), as well as the chaperones Hsp104 and Ssa1 and the autophagy regulator Atg1, to see whether any would impact on the formation of inclusions. We used NaN₃ for induction because the level of inclusion formation could be manipulated by varying the concentration (see, e.g., Fig S4B). With these assays, we found that although deletion of chaperones or autophagy genes did not affect the level of inclusion formation, deletion of the ER–mitochondrial ERMES subunits MDM34 and MMM1 (Kornmann et al,

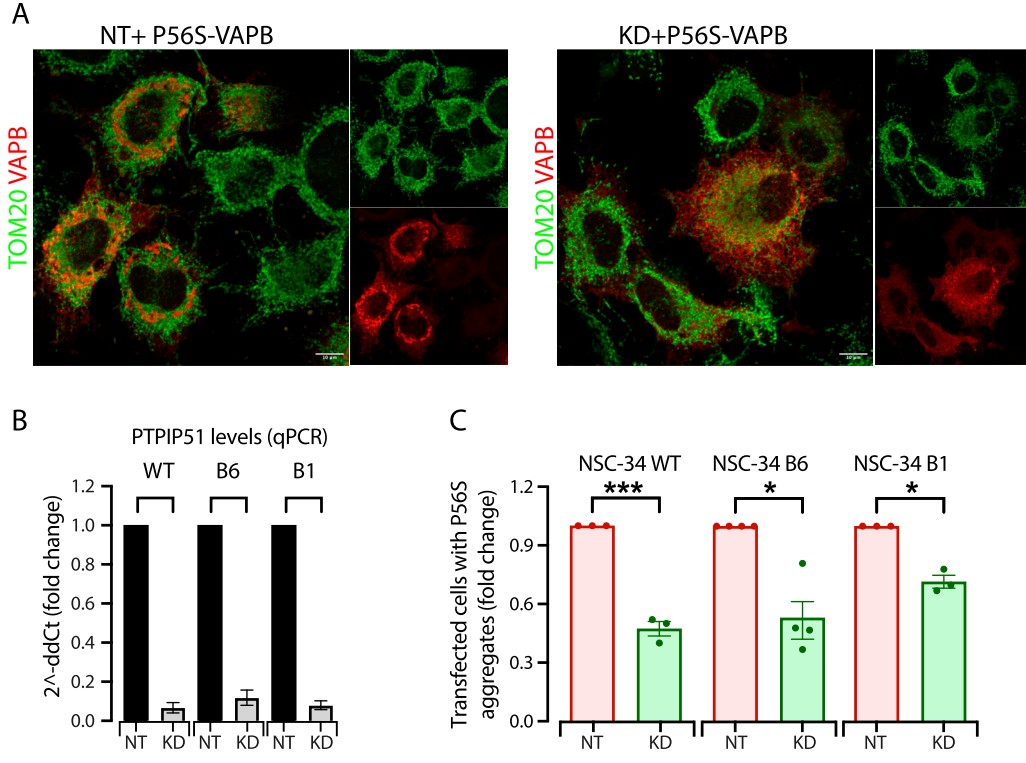

**Figure 7. Down-regulation of PTPIP51 reduces P56S-VAPB aggregate formation in mammalian cells.**
**(A)** Representative immunofluorescence images of NSC-34 B1 cells treated with non-targeting (NT) or PTPIP51 (KD) siRNA and then transfected with a P56S-VAPB–expressing plasmid. The cells were immunostained for VAPB and TOM20. Scale bars, 10 $\mu$m. **(B)** qRT–PCR analysis of PTPIP51 expression after treatment with PTPIP51 siRNA expressed as a fold change with respect to non-targeting (NT) treated WT NSC-34 cells, or the B1 and B6 clones. Mean ± SD, N = 3. **(C)** Quantification of the transfected cells that form P56S aggregates, expressed as a fold change with respect to non-targeting (NT) treated cells. N = 6, n > 100. Statistical significance between non-silenced (NT) and PTPIP51-KD (KD) was calculated by a ratio paired $t$ test. *$P$ < 0.05, ***$P$ < 0.001.
Source data are available for this figure.

2009) reduced the sensitivity to inclusion induction by NaN$_3$, albeit weakly (Fig 6A). In contrast, and more strikingly, deletion of the PM–mitochondrial tether *NUM1* (Lackner et al, 2013) rendered the cells hypersensitive to inclusion induction (Fig 6A). Even a low concentration of NaN$_3$ (0.05 mM), which does not result in inclusion formation in P56S-VAPB-mGFP–expressing cells with a WT genomic background, induced inclusions in the *num1Δ* background (Fig 6B–D).

Num1 is a component of the mitochondria–ER–cortex anchor, which tethers mitochondria to the cell cortex (Lackner et al, 2013) and which is also a direct interacting partner of Scs2 at the PM-associated ER (Chao et al, 2014). Deletion of *NUM1* leads to a defect in mitochondrial respiration (White et al, 2022) and repositions mitochondria in a centralized location (Fig S5A; Klecker et al, 2013; Lackner et al, 2013). Notably, ERMES retains the centralized localization in untreated *num1Δ* cells (Fig S5B). Upon NaN$_3$ treatment, ERMES remains associated with mitochondria both in WT cells where mitochondria are fragmented (Fig S5C) and in *num1Δ* cells where mitochondria are known to be resistant to NaN$_3$-induced fragmentation (Klecker et al, 2013) (Fig S5B).

Thus, although the association between ERMES and mitochondria does not seem to be affected by treatment that induces inclusion formation, their function might be compromised. Indeed, ERMES deletion also decreased the hypersensitivity of the *num1Δ* cells (Fig 6C and D).

## Disruption of mitochondrial contacts reduces inclusion formation in mammalian cells

We then asked whether disruption of ER–mitochondrial contacts in mammalian cells affected inclusion formation. The ERMES complex does not exist in mammals, and multiple tethering factors have been shown to establish ER–mitochondrial contact sites. We focused on PTPIP51, an interactor of VAPB that also acts as a tether between the ER and mitochondria (De Vos et al, 2012). This direct interaction of VAPB with mitochondria via PTPIP51 suggests the possibility that ER–mitochondrial contacts could have a role in disease progression because VAPB-P56S was shown to have a greater affinity for PTPIP51, to increase ER–mitochondria association (De Vos et al, 2012), and to perturb anterograde mitochondrial axonal transport by disrupting calcium homeostasis (Mórotz et al, 2012). PTPIP51 is unlikely to fulfill ERMES's function based on its very different structure and properties, although the membrane tethering and phospholipid transfer function of PTPIP51 suggest that it might act, at least in part, as a functional counterpart of the ERMES complex (Yeo et al, 2021). Indeed, knockdown of PTPIP51 in the motoneuron-like cell line NSC-34 was shown to cause a significant reduction in ER–mitochondrial contacts (Stoica et al, 2014).

PTPIP51 expression was silenced in NSC-34 cells and in the two NSC-34 cell lines mentioned above (B6 and B1) that show reduced VAPB expression. The formation of inclusions was significantly reduced after KD of PTPIP51 in all three cell lines (Fig 7A–C), consistent with the results in yeast where reducing ER–mitochondrial contacts reduces aggregate formation.

# Discussion

There have been many studies on VAPB and its role in ALS8 that have used cell systems, IPSC-derived neurons, and animal models, all of which have provided invaluable information regarding VAPB function and ALS8 pathogenicity. One issue that has been problematic in studying the mutant VAPB protein is its propensity to aggregate and form inclusions when overexpressed. Although yeasts are obviously a long way from mammalian cells and neurons, we believe that the yeast model presented here provides an in vivo situation to study the biophysical properties of the mutant protein. In addition to VAPB having structural and functional similarities to its yeast ortholog Scs2, the P56S-VAPB inclusions in yeast cells have characteristics of their mammalian counterparts. The expression setup we developed has revealed properties of the protein that promise to be relevant for understanding its behavior in a disease context. Recently, a similar yeast model for ALS8 was generated, although in this case, cells deleted of both yeast VAPB paralogues, that is, SCS2 and SCS22, and expressing either mutant SCS2 or the human P56S-VAPB were used (Stump et al, 2023). Unlike our approach, Stump and colleagues used their yeast model not to address the issue of P56S-VAPB inclusions but to assess the ability of P56S versus WT-VAPB to rescue the altered ER morphology and ER stress induced by the combined deletion of SCS2 and SCS22. They found that although the WT human VAPB protein rescued both phenotypes, the P56S mutant partially rescued ER stress but not ER morphology.

We report that the P56S-VAPB protein does not cause inclusion formation in yeast except under particular circumstances. The data in this work point to mitochondrial damage as being a likely trigger. The observation that P56S-VAPB inclusions occur in a subpopulation of stationary-phase cells is of particular interest in this regard. The generation of two different cell types in the stationary phase is a programmed developmental step of yeast that leads to cellular differentiation (Davidson et al, 2011; Sagot & Laporte, 2019) in which one population (non-quiescent) has damaged mitochondria, whereas the other (quiescent) has functional mitochondria (Laporte et al, 2018). The former, but not the latter, form inclusions. It is of interest that calorie restriction was found to reduce P56S-VAPB inclusion formation in stationary-phase cells because dietary restriction enhances mitochondrial functions and attenuates age-related declines in mitochondrial function (Ruetenik & Barrientos, 2015), illustrating that, in principle, it is possible to intervene and prevent the formation of mutant VAPB inclusions.

However, it does not seem to be mitochondrial dysfunction per se that induces inclusion formation because the num1Δ mutant, despite reduced respiratory capacity, did not form inclusions

spontaneously but only after applying mitochondrial damage or in the stationary phase. Rather, the fact that disrupting ER–mitochondrial contact sites reduces inclusion formation after the application of a mitochondrial insult suggests that there may be the transmission of a deleterious signal to the ER. The observation that the expression of P56S-VAPB in yeast cells leads to a disruption of ER morphology (Stump et al, 2023) could render the mutant protein more prone to aggregation upon an additional insult. The hypersensitivity of the num1Δ strain, which disrupts PM–mitochondrial contacts, after mitochondrial insult might be due to the concentration of the ER–mitochondrial contact sites to the centralized mitochondria, because this hypersensitivity is reduced upon elimination of ERMES. Interestingly, it has been reported recently that the diffusion landscape of VAPB at contact sites is a crucial component of ER–mitochondrial contact site homeostasis. A portion of P56S-VAPB molecules is trapped in multiple small subdomains disrupting their normal diffusion at the contact site, which prolongs its permanence at the contacts and, potentially, the exposure to mitochondria-derived pro-aggregation signals (Obara et al, 2024).

The results thus indicate that P56S-VAPB, an ER protein, aggregates upon mitochondrial damage and that inclusion formation is attenuated by disrupting ER–mitochondrial contact sites, suggesting that interorganelle signaling could be at the basis of this phenomenon. What this deleterious event might be is the subject of ongoing studies but must take into consideration that the aggregation-prone domain is exposed to the cytosol. ER–mitochondrial contacts are sites of active exchange of lipids and of redox signaling, and one can hypothesize that oxidative bursts within the nanoscale environment of interorganellar contact sites mediate retrograde signaling toward the ER that generates the pro-aggregation oxidative environment (Booth et al, 2016, 2021).

A further finding relevant for understanding ALS8 etiology was the observation that the WT protein slowed down inclusion formation in diploid cells that expressed both WT and mutant alleles at endogenous levels, thus mimicking the disease situation, and a similar effect was observed in NSC-34 cells where there was an inverse correlation between WT protein expression levels and inclusion formation. This could be explained by the dynamic interaction between VAPB molecules, which are known to dimerize/oligomerize (Kim et al, 2010). The mutant protein has been proposed to cluster, generating tightly apposed ER cisternae held together by interactions between the mutant cytosolic MSP domains (Fasana et al, 2010; Papiani et al, 2012), a phenomenon that was replicated in our yeast model and supported by the liposome–aggregation assay performed with the WT and P56S MSP domain anchored to the liposome. Our results are in line with those reported by Kim et al (2010) who compared the oligomerization properties of WT and P56S-VAPB proteins. They found that the oligomerization of the WT-VAPB is independent of its MSP domain but requires the coiled-coil domain, whereas the P56S mutation, possibly by inducing conformational changes within the MSP domain, which we report are further promoted by an oxidative environment, facilitates its propensity to aggregate. We propose that the oligomerization of the WT-VAPB involving mainly the coiled-coil domain (near the TM domain) mediates an "in cis" interaction of WT-VAPB, whereas the oligomerization of P56S-VAPB involving the cytosolic MSP domain

mediates an "in trans" interaction and promotes the tethering of facing ER membranes, leading to inclusion formation.

The co-presence of the WT protein could slow down this process by associating with and retarding the oligomerization of mutant proteins, although in a dynamic situation, the mutant protein will eventually overcome this equilibrium if it forms insoluble clusters, a scenario that is supported by the observation of an impairment in the dynamic restructuring of ER–mitochondrial contact sites in the presence of the P56S mutation, which could alter the dimerization/lateral aggregation of VAPB within the contact site (Obara et al, 2024).

These findings could have relevance to the etiology of ALS8, as well as other late-onset neurodegenerative diseases. The presence of the mutation itself may not be deleterious in "young" cells (as a surrogate, log-phase yeast cells), but only becomes deleterious when associated with a secondary event, for example, mitochondrial damage in aged cells (as a surrogate, NQ stationary-phase yeast cells). This may occur only in a particular cell type (neurons or NQ yeast cells) but not in other cells where the two phenomena do not co-occur (non-neuronal cells or Q yeast cells), when the vulnerable cell type passes a threshold because of the combination of genetic predisposition along with aging and/or environmental factors.

In such a scenario, the motoneurons that are damaged in ALS would represent the vulnerable cell type. They are enriched for mitochondria at the neuromuscular junction (Altman et al, 2019) and have low levels of cytosolic calcium-buffering proteins that enable rapid calcium signaling to produce high-frequency spiking (Siklós et al, 1998; Vanselow & Keller, 2000; Jaiswal & Keller, 2009). Thus, motoneurons affected in ALS are particularly prone to excitotoxicity, cellular damage because of calcium overload, and cell stress, making them exceptionally vulnerable to disruption of mitochondrial function. A stressor–threshold model of selective neuronal vulnerability and of the role of neuronal vulnerability in disease is consistent with a large body of observations in patients and in animal models (Saxena & Caroni, 2011).

Mitochondrial dysfunction has been widely cited as a contributing factor in neurodegenerative diseases, including ALS (for reviews, see Grimm & Eckert, 2017; Smith et al, 2019). Indeed, aberrant mitochondrial morphology and impaired mitochondrial function have been reported in postmortem samples from ALS patients (Borthwick et al, 1999; Wiedemann et al, 2002). A vicious cycle of events triggered in a mutated protein background caused by interorganelle signaling between mitochondria and the ER, as well as other organelles, could then escalate into the disease phenotype, contributing to the progressive nature of ALS and other late-onset diseases. Indeed, structural damage to mitochondria in early disease stages in in vivo models of ALS has been proposed as an upstream source of degeneration rather than a consequence (Vande Velde et al, 2011; Magrané et al, 2014). Furthermore, another contributing factor to the progressive nature of the disease is the observation that the WT protein may suppress inclusion formation, and this protective mechanism is only overcome as the aggregation process, stimulated by increasing cell damage, becomes overwhelming. This situation could be exacerbated by a failure of proteostasis to clear the mutant protein, which is unstable in cell culture models (Papiani et al, 2012; Genevini et al, 2014), allowing it to accumulate to harmful levels.

The fact that mitochondrial dysfunction in many neurodegenerative diseases appears as an age-related phenomenon could potentially explain why P56S-VAPB inclusions were not observed in ALS8 patient IPSC-derived neurons (Mitne-Neto et al, 2011), in that these are not aged cells. Motoneurons derived from ALS patient IPSCs carrying FUS mutations did not show any mitochondrial morphology or metabolic dysfunction (Vandoorne et al, 2019). Furthermore, the current findings might be relevant for other ALS-causing genes. Mutant SOD1, associated with familial and sporadic ALS, was found to aggregate in aged yeast cells but not in exponentially growing cells (Brasil et al, 2019). SOD1, TDP-43, and FUS ALS all disrupt ER–mitochondrial contacts, and in the case of TDP-43 and FUS, this may involve altered VAPB-PTPIP51 association (Stoica et al, 2014, 2016).

We propose that it is not simply the mutant protein that damages mitochondria but, as described in this work, that damage to mitochondria influences the mutant protein, initiating a series of events leading to disease progression. Different mechanisms have been proposed to account for the dominant inheritance observed in ALS8, from gain of function to loss of function or haploinsufficiency (Borgese et al, 2021a, 2021b). Regardless of which mechanism turns out to be the correct one, we would argue that identifying a primary event that triggers protein aggregation preceding such dysfunctional mechanisms provides a therapeutic window for intervention.

# Materials and Methods

## Yeast methods

### Plasmid construction

The starting human VAPB cassette was a GeneArt Strings DNA fragment from Thermo Fisher Scientific, encoding the entire human VAPB cDNA (codon-optimized for expression in yeast) with monomeric GFP (mGFP, A206K mutant, Zacharias et al, 2002) inserted before the VAPB transmembrane domain (TMD). This was flanked by the *SCS2* promoter and the *SCS2* terminator with *BamHI* and *SalI* cloning sites in the order: *BamHI*-SCS2p-VAPB(N)-mGFP-VAPB(TMD)-SCS2t-*SalI* (see Fig 1A and Table S1A). The cassette was PCR-amplified using oligos VAPB.BamHI.Fwd and VAPB.SalI.Rev (Table S1B) and cloned into *BamHI*/*SalI* vectors pRS41N, pRS41K, and pRS41H (Table S2) that contain the dominant drug resistance markers nourseothricin (natNT2), geneticin (kanMX4), and hygromycin B (hphNT1), respectively (Taxis & Knop, 2006). The P56S mutation and untagged WT and P56S versions were produced using QuikChange II Site-Directed Mutagenesis Kit (Agilent). The mGFP was exchanged with RUBY2 using pFA6a-link-yomRuby2-Kan (plasmid #44953; Addgene, Lee et al, 2013) as a template and the Polymerase Incomplete Primer Extension method (Klock & Lesley, 2009). All clones were verified by sequencing.

To generate the mitochondrial reporter pRS41N-MTS-Cherry, an *XbaI*/*EcoRI* fragment containing a MTS fused to mCherry was taken from the vector p404-MTS-Cherry-GFP1-10 (plasmid #91958; Addgene, Ruan et al, 2017) and cloned into a pRS41N plasmid derivative containing the GAPDH promoter and CYC terminator (pRS41N-Gap-Cyc, this study). pRS41K-MTS-Cherry was obtained by cloning the entire *SacI*/*SalI* cassette from pRS41N-MTS-Cherry into pRS41K.

pRS41N-MTS-mGFP was obtained by replacing Cherry with the mGFP sequence from pRS41H-VAPB-mGFP using oligos mGFP.Bam.Fwd and mGFP.Eco.Rev (see Table S1B).

pRS41N-mGFP was constructed by amplifying mGFP using oligos mGFP.Xba.Fwd and mGFP.Eco.Rev (Table S1B) with pRS41H-VAPB-mGFP as a template and cloning *XbaI*/*EcoI* into pRS41N-Gap-Cyc.

pRS41N-MDM34-mGFP was constructed by first amplifying the *MDM34* gene including 660 bp of the promoter from yeast genomic DNA with oligos Mdm34.SacI.Fwd and Mdm34.XbaI.Rev and cloning *SacI*/*XbaI* into pRS41N-Gap-Cyc, thus removing the GAPDH promoter and producing pRS41N-Mdm34. The mGFP fragment was digested *XbaI*/*EcoRI* from plasmid pRS41N-mGFP and cloned into pRS41N-MDM34 producing C-terminally tagged Mdm34.

### PCR amplification for gene replacement

Amplicons were amplified from plasmids such that the coding sequence or promoter of the desired replacement sequence was amplified by PCR primers with overhangs homologous to the targeted genomic flanking region (Table S1B and see below). The following conditions were used: 98°C, 2 min; 98°C, 30 s; 62°C, 30 s; 72°C, 2 min 40 s, for 10 cycles; then 98°C, 15 s; 62°C, 30 s; 72°C, 2 min 40 s, for 25 cycles with a 10-s extension every cycle; and finally 72°C, 10 min. Amplicons were purified from gels using the Sigma-Aldrich gel extraction kit.

### Yeast strains

Yeast strains used in this study are listed in Table S3. The VAPB strains were created by replacing the yeast *SCS2* gene with the hVAPB cassette (containing mGFP, RUBY, or untagged) using gene replacement via homologous DNA recombination in both the BY4741 (mat a) and BY4742 (mat alpha) strains. Amplicons were obtained from the various plasmids using oligos VAPB.Prom.HI.Fwd and VAPB.TEFp.HI.Rev (TEFp refers to the TEF promoter that drives the antibiotic resistance in pRS41N, pRS41K, and pRS41H). Diploids were obtained by mating VAPB strains having complementary selection markers and differentially fluorescent tagged/untagged alleles. Other expression cassettes were also introduced into the genome using gene replacement via homologous DNA recombination with oligos YFR.MDM34.Rev/YFR.Tefp.Fwd (Table S1B) and pRS41N-MDM34-mGFP as a template to insert MDM34-mGFP into the putative ORF locus YFR054C. The MTS-Cherry was amplified from pRS41K-MTS-Cherry using oligos YMR.TEFp.Fwd/YMR.GAPp.Rev and inserted into the putative ORF locus YMR082C.

### Yeast transformation

1 ml of an overnight culture was spun, the supernatant was removed, and ~1 µg of DNA amplicon, or 1 µl of plasmid miniprep, was added to the cell pellet with 10 µl denatured salmon sperm DNA (Invitrogen). The cells were resuspended in 500 µl PEG/LiAc (40% PEG 3350, 0.1 M LiAc, 10 mM Tris-Cl, pH 7.5, 1 mM EDTA) and incubated at room temperature for 30 min. After heat shock at 42°C for 10 min, 55 µl ethanol was added and mixed, and the heat shock continued for 5 min. Cells were spun (4,700*g*, 1 min), washed once, and then plated on selection medium. Gene replacement was verified by PCR

analysis on genomic DNA isolated from transformants (Yeast DNA Extraction Kit; Thermo Fisher Scientific).

### Growth conditions

Cells were grown in YPD (1% yeast extract, 2% peptone, 2% glucose, with the addition of 2% agar for solid media) as preinoculation cultures. These were used to inoculate synthetic complete (SC) medium (0.67% yeast nitrogen base to which all amino acids were added) containing either 2% glucose (SD) or 2% galactose (SG). Inositol-reduced medium was SD but prepared using Yeast Nitrogen Base without amino acids and inositol (CYN3701; Formedium) to which inositol (DOC0198; Formedium) was added to 1 mM.

Cells were grown in YPD for 3 d for observation of cells in the stationary phase. Calorie restriction used YPD with 0.5% glucose.

Except for time course experiments, induction of inclusion formation was performed in SD or SG medium for 60 min. Cells were grown to an OD~1.0 before treatment. For carbon source removal, the cells were spun, washed briefly, and resuspended in SC medium. Antimycin A was added to galactose-depleted cells at a concentration of 10 µM. 2-Deoxyglucose treatment was in SD with 0.2% (11.11 mM) glucose plus 33.33 mM 2-DOG. The final concentration of the following compounds was used: $H_2O_2$ (1.5 mM), CCCP (50 µM), N-acetylcysteine (100 mM), DTT (8 mM).

### Cell fractionation

Cells were grown in YPD for 3 d to the stationary phase. Non-quiescent (NQ) and quiescent (Q) cells were isolated by Percoll gradient centrifugation as described by Lee et al (2016).

### ATP measurements

ATP was measured as described by Ashe et al (2000) using Enliten luciferin/luciferase reagent (Promega) and GloMax 96 Microplate Luminometer (Promega) with a 10-s integration period.

### Triton X-100–resistant aggregates

Cells were grown in 50 ml SD overnight to OD 1.2, treated or not for 60 min with 0.1 mM NaN$_3$, collected by centrifugation (1,500*g*, 10 min), washed briefly with water, and resuspended in 500 ml extraction buffer (20 mM Hepes/NaOH, pH 7.3, 150 mM NaCl, 5 mM EDTA 1X protease inhibitor cocktail, 1 mM PMSF). One half-volume of glass beads was added, and the cells were lysed on a bead beater for 15 min, 4°C. After a low-speed spin (500*g*, 5 min), the supernatant was taken (total lysate) and protein concentration was determined (Bradford assay). The same total amount of protein was taken for each sample and adjusted to the same volume. Samples were centrifuged at 21,000*g*, 15 min, 4°C, the supernatant was collected (S21), and the pellet was resuspended in the same volume of lysis buffer (P21). Triton X-100 was added to 1% and incubated at 4°C, 30 min, with gentle agitation. After centrifugation at 21,000*g*, 15 min, 4°C, the supernatant was collected (S21T) and the pellet was resuspended in the same volume of lysis buffer (P21T). An equal volume of each fraction was taken, one volume of 4x SDS loading

buffer was added, and the samples were heated to 65°C, 10 min, and then centrifuged at 21,000$g$, 5 min.

## Total cell proteins

Total cell proteins were prepared by resuspending cell pellets in 400 $\mu$l of cold NaOH (0.15 M), vortexing briefly, and leaving on ice for 5 min. The cells were centrifuged at 4,700$g$, 4°C, 3 min, the supernatant was removed, and the pellets were resuspended in 50 $\mu$l of 2xSDS loading buffer for each 1 × 10$^7$ cells. After a brief vortex, the cells were heated at 65°C, 10 min, and then centrifuged at 21,000$g$, 5 min.

## Western blot analysis

We systematically assessed by WB the amount of the expressed WT and P56S proteins in the different strains and under the different experimental conditions and verified that they were expressed at comparable levels.

Protein samples were separated on precast 4–20% Mini-PROTEAN TGX gels (Bio-Rad) and blotted to nitrocellulose membranes. Primary antibodies used were anti-VAPB antibody 1:5,000 (rabbit, in-house), anti-Kar2 1:2,000 (rabbit, y-115, sc-33630; Santa Cruz Biotechnology), and anti-PGK1 1:10,000 (mouse mAb 22C5D8; Thermo Fisher Scientific).

## Fluorescence microscopy

Live-cell imaging was performed on a DM5000 B fluorescence microscope (Leica) for GFP-, Ruby-, and mCherry-tagged proteins. Cells were immobilized on concanavalin A (Sigma-Aldrich)–coated slides before imaging. Images were captured with Leica Application Suite software. To visualize mitochondrial membrane potential, cells were incubated with 100 nM MitoTracker Red CMXRos (Molecular Probes) for 30 min at 30°C, washed once in YNB, and resuspended in YNB before microscopy. To visualize vacuolar pH, cells were stained with quinacrine (Sigma-Aldrich) by cooling cells on ice for 5 min, centrifuging, and resuspending in 100 $\mu$l YPD containing 100 mM Hepes, pH 7.6, and 200 $\mu$M freshly prepared quinacrine. After incubation for 5–10 min at 30°C, the cells were cooled on ice for 5 min and washed twice in cold 100 mM Hepes, pH 7.6, and 2% glucose before microscopy. Bafilomycin was used at 5 $\mu$M for 30 min before staining. Brightness and contrast were adjusted using Fiji (ImageJ2).

## CLEM

For CLEM, yeast cells expressing P56S-VAPB-mGFP were grown in SD-Trp at 30°C to the log phase. 0.015% NaN$_3$ was added to the culture for 90 min before cells were pelleted by vacuum filtration and cryo-fixed in aluminum planchettes (Wohlwend) using an HPM100 high-pressure freezer (Leica Microsystems). Freeze substitution (FS) and embedding in Lowicryl HM20 were done as described previously (Kukulski et al, 2011; McDonald & Webb, 2011) except that samples were shaken on dry ice for 3 h during FS before being placed into the AFS2 (Leica Microsystems), and the uranyl acetate concentration in the FS solution was less than 0.03%. CLEM

on resin sections was performed as described previously (Ader & Kukulski, 2017; Hoffmann et al, 2019) except that for correlation, the centers of cells visible in fluorescence images and montaged electron microscopy images (pixel size 3.1 nm) of the area of interest were used instead of fiducial markers, using the MATLAB (MathWorks) scripts as described previously (Kukulski et al, 2011). Electron tomography was performed in scanning transmission EM mode on a TF20 microscope (FEI) as described before (Hohmann-Marriott et al, 2009; Ader & Kukulski, 2017; Hoffmann et al, 2019). Dual-axis tilt series (Mastronarde, 1997) were acquired at ±60° with 1° increment, at 1.1-nm pixel size, using SerialEM (Mastronarde, 2005). Tomogram reconstructions and segmentation were done in IMOD (Kremer et al, 1996). Segmentation model rendering was done in UCSF ChimeraX (Meng et al, 2023). For visualization in the figure, median filtering was applied to the tomogram. Twelve P56S-VAPB-mGFP inclusions were targeted by CLEM. Of the resulting tomograms, three were of too poor quality for interpretation. The remaining nine tomograms all contained paired ER cisternae at the position of the GFP signal.

## Mammalian methods

### Cell culture

Neuroblastoma x Spinal Cord-34 (NSC-34) cells were maintained in DMEM supplemented with 1% Na$^+$ pyruvate, 10% FBS, 100 U ml$^{-1}$ penicillin, and 100 $\mu$g ml$^{-1}$ streptomycin (Thermo Fisher Scientific). Cell lines NSC-34 B6 and B1 were generated as previously described (Genevini et al, 2019).

## VAPB constructs

Human VAPB (Gene ID 9217) was amplified by PCR with primers flanked by ATTL1/2 sites for the Gateway system and recombined in two steps into the PCDNA5/FRT/TO vector (Thermo Fisher Scientific).

P56S-VAPB was generated from VAPB with the QuikChange site-directed mutagenesis kit (Agilent) according to the manufacturer's instructions, using the following primers: 5'-GTAGGTACTGTGT-GAGGCCCAACAGCGGAATCATC-3', 5'-GATGATTCCGCTGTTGGACCTCA-CACAGTACCTAC-3'.

## Transfection

NSC-34 cells (WT, B6, and B1) were plated in six-well plates (for WB analysis) or 24-well plates (for immunofluorescence analysis) and transfected with WT or P56S-VAPB plasmids for 24 h, using Lipofectamine LTX and PLUS$^{TM}$ Reagent (Thermo Fisher Scientific), according to the manufacturer's instructions.

## RNA interference

NSC-34 cells (WT, B6, and B1) were seeded in six-well plates and treated with 50 nM siRNA (PTPIP51 or VAPA) or non-targeting siRNA for 96 h using Lipofectamine RNAiMAX (Thermo Fisher Scientific) for forward transfection. After 72 h, cells were plated in 24-well plates and transfected with WT or P56S-VAPB plasmids using

Lipofectamine LTX and PLUS Reagent (Thermo Fisher Scientific), according to the manufacturer's instruction.

The siRNA sequences for mouse PTPIP51 were 5′-GGAUGA-CAACGCUGGCAAAGGGU-3′ and 5′-AGGUUAUACAACAGCCAACGCGG-3′. The siRNA sequences for mouse VAPA were 5′-CCUGAU-GAAUUAAUGGAUU-3′ and 5′-CUUUGAUUAUGAUCCGAAU-3′.

## Immunofluorescence analysis

Immunofluorescence analysis was performed as previously described (Venditti et al, 2019). The anti-VAPB antibody (cod. 66191-1-Ig) and the anti-TOM20 antibody (cod. 11802-1-AP) were from ProteinTech.

## Real-time PCR

The real-time qPCR was performed using LightCycler 480 II System (Roche) in 96-well plates, manually set up in triplicates using oligos:

Primer qRT-PCR PTPIP51+ : 5′-AAGAAAGGAGATGAGAGTGCTG-3′.
Primer qRT-PCR PTPIP51– : 5′-GCTTTGTCCACATGTTCCTTG-3′.

## Recombinant protein preparation and analysis

SUMO-His-tagged MSP domains were expressed in Rosetta (DE3) *E. coli* and purified using Ni-NTA agarose (QIAGEN). 6 μg of each protein was diluted in sample buffer with or without reducing agents and loaded on 12% polyacrylamide gel without boiling the samples. Proteins transferred to a nitrocellulose filter were visualized by Ponceau staining.

## Liposome preparation and MSP domain self-association assay

Liposomes were prepared from lipids DOPC, DOPS, and DOGS-Ni-NTA (molar ratio 91:5:4), dried to a film, hydrated with 50 mM Hepes, pH 7.4, and 120 mM KAc, and extruded using a poly-carbonate filter with a pore size of 100 nm. MSP-SUMO-His domains from WT or P56S-VAPB (5 μM) were incubated with 1 mM of the prepared liposomes and incubated at 4°C for 1 h. Liposome aggregation was assessed by measuring the absorbance at 410 nm every 2 min over 90 min at 30°C using Neo2 Microplate Reader (Biotech).

## Pull-down assay

4 μg of WT or P56S MSP-SUMO-His proteins was incubated overnight at 4°C with 4 μg of untagged WT or P56S MSP proteins (SUMO-His-tagged proteins treated with SUMO protease) in 300 μl of binding buffer (25 m Tris–HCl, pH 7.4, 150 mM NaCl, 0.2% Triton X-100, and 0.5 mg/ml BSA fatty acid free) supplemented with Ni-NTA agarose and 1 mm DTT (where indicated). Beads were washed once in binding buffer and then four times in binding buffer without BSA. After resuspension in gel loading buffer, supernatants were analyzed by 12.5% SDS–PAGE.

## Western blot analysis

NSC-34 cells (WT, B6, and B1) were lysed in 25 mM Tris–HCl, pH 7.4, 150 mM NaCl, 1 mM EDTA, and 0.5% Triton X-100, supplemented with protease inhibitors. 30 μg of cell lysates was used for Western blot analysis. The anti-VAPB antibody was used as previously described (Genevini et al, 2019).

## Quantifications and statistical analysis

Detailed statistics including the number of cells analyzed, mean value, SD, and *P*-value are indicated in each figure legend.

# Supplementary Information

# Acknowledgements

We thank Manuel Muniz (University of Seville, Spain) for the yeast total protein extraction procedure and Andrea Guarino for drawing the graphical abstract. This work was supported by the Fondazione Telethon ETS, the Italian Association for Cancer Research (grant IG2023_29210), the Italian Ministry of University and Research (PRIN2020PKLEPN; PRIN 2022 F2YJNK; PRIN 2022 PNRR 20223NAJ5), #NEXTGENERATIONEU (NGEU), and by the Ministry of University and Research (MUR), National Recovery and Resilience Plan (NRRP), project MNESYS (PE0000006)—A multiscale integrated approach to the study of the nervous system in health and disease (DN. 1553 11.10.2022)—FIS00002484 to MA De Matteis. W Kukulski was supported by the Medical Research Council, as part of United Kingdom Research and Innovation (also known as UK Research and Innovation) under award MC_UP_1201/8. R Venditti acknowledges the support from AIRC (MFAG 2020_25174) and the Italian Ministry of University and Research (PRIN, 20222 MAWZP). We thank the EM facility of the MRC LMB for support. The icons used in the graphical abstract were taken from the online platforms Alamy and BioRender.

## Author Contributions

C Wilson: data curation, formal analysis, investigation, methodology, and writing—original draft, review, and editing.
L Giaquinto: data curation, validation, and methodology.
M Santoro: methodology.
G Di Tullio: methodology.
V Morra: investigation and methodology.
W Kukulski: data curation and methodology.
R Venditti: conceptualization and data curation.
F Navone: conceptualization.
N Borgese: conceptualization and data curation.
MA De Matteis: conceptualization, resources, supervision, funding acquisition, and writing—original draft, review, and editing.

## Conflict of Interest Statement

The authors declare that they have no conflict of interest.

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
