## [Reviewer comments · Life Science Alliance]

Life Science Alliance

A role for mitochondria-ER crosstalk in amyotrophic lateral sclerosis 8 pathogenesis

Cathal Wilson, Laura Giaquinto, Michele Santoro, Giuseppe Di Tullio, Valentina Morra, Wanda Kukulski, Rossella Venditti, Francesca Navone, Nica Borgese, and Maria Antonietta De Matteis

DOI: <https://doi.org/10.26508/lsa.202402907>

Corresponding author(s): Maria Antonietta De Matteis, Telethon Institute Of Genetics And Medicine

Review Timeline:	Submission Date:	2024-06-25
	Editorial Decision:	2024-08-09
	Revision Received:	2024-11-28
	Editorial Decision:	2024-12-24
	Revision Received:	2025-01-07
	Accepted:	2025-01-08

Transaction Report:

August 9, 2024

Re: Life Science Alliance manuscript #LSA-2024-02907-T

Dr. Antonella De Matteis
Telethon Institute of Genetics and Medicine
Via Campi Flegrei 34
Pozzuoli, Naples 80078
Italy

Dear Dr. De Matteis,

Thank you for submitting your manuscript entitled "The role of ER-mitochondrial contact sites in amyotrophic lateral sclerosis 8 (ALS8) pathogenesis" to Life Science Alliance. The manuscript was assessed by expert reviewers, whose comments are appended to this letter. We invite you to submit a revised manuscript addressing the Reviewer comments.

Thank you for this interesting contribution to Life Science Alliance. We are looking forward to receiving your revised manuscript.

Sincerely,

B. MANUSCRIPT ORGANIZATION AND FORMATTING:

Reviewer #1 (Comments to the Authors (Required)):

Comments to the authors:

The manuscript entitled "A role of ER-mitochondrial contact sites in amyotrophic lateral sclerosis 8 (ALS8) pathogenesis" by C. Wilson et al., describes the observations that can be made when human VAPB, the protein mutated in ALS8 is expressed in the model organism *S.cerevisiae*. To achieve "physiological" levels of VAPB the gene encoding the yeast paralogue SCS2 was replaced by WT and mutant VAPB by homologous recombination. The main message of the study is that in yeast the mutated VAPB favors inclusion/aggregate formation at the ER that is evoked by a mitochondrial aberration. The authors claim that the mitochondrial changes are relayed to the ER by mitochondrial-ER contact sites. The observations made are quite interesting and the study is well executed. However, I have some concerns that should be addressed.

1) "Physiological expression levels of VAPB in yeast". In my opinion it is rather difficult to determine such an expression level for a paralogue from another organism. Not only expression by a certain promoter determines protein levels but also e.g. protein stability or posttranslational modifications.

The authors should construct a yeast strain that contains Scs2p with a GFP-module similarly inserted next to the TMD as done for VAPB. Using these cells at least similar protein-levels of Scs2p and VAPB can be shown.

The functionality assay (FigS1A) is not really convincing since although WT-VAPB shows some rescue of growth overall (approx. 50-60%) in comparison to *scs2d* cells, the single colonies are rather small in comparison to BY4742 WT, reminiscent of petite colonies formed by yeast cells that have respiratory/mitochondrial defects. To show loss of function of the P56S mutant the authors should show by western blot that the cells of FigS1A have comparable VAPB (WT or P56S) expression. By this a lower P56S expression/stability in the cells responsible for the lack of rescue, can be excluded.

How highly conserved is VAPB to SCS2 and SCS22?! Would it be possible to introduce the P56S mutation in Scs2 or Scs22 and mimic/observe the inclusion/aggregate with the yeast protein?!

2) Protein steady state levels should be determined by Western Blot with antibodies against VAPB or GFP (if the P56S mutation alters ab affinity) for all experiments where the WT VAPB is compared to the mutant for forming the aggregates/inclusions.

3) For all conditions where the inclusions were observed it should be confirmed, that one is dealing with the same type of protein inclusions, e.g. the TX-100 resistance assay.

4) Figure 2B: The same experiment should be performed with WT VAPB to see what the ER looks like without the inclusion formation.

5) Figure 2C: Why is a higher molecular weight adduct (via disulfide bridges) forming under reducing conditions of the cytosol (the N-terminus of VAPB faces the cytosol). Could this be during the lysis of the cells?

6) Are the punctate structures observed in case of the P56S mutant ER-bound or protein aggregates in the cytosol. To this end the authors should show by cell fractionation experiments that WT and mutant VAPB can be found in the ER (microsome) fraction.

7) Figure 3 A: The combination with a mutant variant should be shown also in log phase

Figure 3B: The WT combination (Fig3A) should also be analyzed and shown under glucose red.

Figure 3C: The WT combination (Fig3A) should also be analyzed and shown under N3- treatment.

8) Figure 6: Since deletion of different mitochondrial morphology genes have different impact on the growth phenotype of yeast cells, which in turn might impact protein expression/stability the expression levels of VAPB-P56S should be compared for all of the mutants.

Also especially in those deletion mutants that are clearly different from WT (ERMES/Num1) the behavior of WT-VAPB should be analyzed.

9) Figure S5: ERMES subunits were reported to show multiple foci on mitochondria in fluorescence-microscopy. Therefore, the uniform cytosolic distribution of Mdm34-GFP is rather unusual. The authors should provide (grayscale) pictures, where the usual ERMES distribution can be observed.

Reviewer #2 (Comments to the Authors (Required)):

The P56S mutation in the MSP domain of VAPB causes a familial form of ALS (ALS8). The mechanism(s) through which this mutation results in disease remains elusive in spite of intense investigation. This condition has dominant transmission, and there has been evidence for both toxic gain of function or haploinsufficiency. It has been reported that VP56S VAPB can form aggregates in cells, most likely due to misfolding, but formation of these aggregates varies with experimental conditions. Moreover, the link of aggregates to disease remains unclear. In this study the authors developed a yeast model to study conditions that lead to P56SVAPB aggregation. They find that mitochondrial dysfunction facilitates aggregate formation and that disruption of ER-mitochondria tethers reduce aggregate formation. Based on these findings they propose that that some signal transmitted to the ER from damaged mitochondria may facilitate aggregate formation. They also report that co-expression of the WT protein also reduced aggregate formation, suggesting a protective effect.

These findings advance knowledge about P56SVAPB pathology and will be of interest in the ALS field. However, many open questions remain and I find the key conclusion "the role of ER-mitochondria contacts in pathology" premature based on the data shown. I suggest to present this idea as a hypothesis, not as a conclusion and to remove it from the title.

Major comments

The authors quote in the introduction the previous study by Stump et al (2023) which capitalized on an approach similar to the one used here to study the impact of the Als8 mutation in yeast. It would be helpful if the authors discussed in more detail the relations of their findings to those of Stump et al.

Evidence that reduction of ER-mitochondria contacts has an attenuating effect on pathology is preliminary. The authors base this conclusion primarily on data involving disruption of the ERMES complex in yeast and of the VAPB - PTPIP51 interaction in mammalian cells. But ERMES (which does not exist in mammals) and the contacts between ER and mitochondria mediated by VAPB and PTPIP51 have very different functions. ERMES does not exist in mammalian cells, and PTPIP51 is unlikely to fulfill ERMES's function based on its very different structure and properties. This should at least be mentioned. Moreover, PTPIP51 directly interacts with VAP. Thus, there may be compounding effects. The strong emphasis on the role of ER-mitochondria contacts should be downplayed, unless this topic is further developed.

Experiments were carried out in yeast cells where the SCS2 gene was replaced by human VAPB. Both in humans and in yeast, there are two VAP genes: VAPA and VAPB in mammals and SCS2 and SCS22 in yeast. In the experiments focused on dimerization, can the author comment on the potential confounding effect of the other isoform?

The MSP domain of VAB can dimerize (PMID: 16004875), as also reported in this study. One would expect that physiological dimerization occurs "in cis", i.e. dimerization of two VAP proteins on the same membrane. The dimerization proposed here to mediate ER aggregation is expected to occur "in trans", on two closely apposed ER membranes. As WT VAP does not induce these appositions, the dimerization generated by P56SVAPB is likely to be mediated by a different type of dimerization. The authors should comment on this.

WT VAPB is reported here to be protective on the formation of aggregates induced by the P56SVAPB mutations. It would be interesting to know if VAPA is also protective.

Reviewer #3 (Comments to the Authors (Required)):

Wilson et al. introduce an interesting series of experiments based on yeast lines in which they have replaced the major cytoplasmic domains of endogenous VAPB ortholog SCS2 with GFP-tagged human VAPB with or without the ALS-causative mutation P56S. They make a number of interesting observations, including that P56S inclusions are only induced under specific growth conditions, that this is dependent on mitochondrial function, and that it is most affected by sites of contact with the endoplasmic reticulum, rather than as one might expect on ATP levels or other obvious metabolic signals.

Although the use of Yeast as a model system certainly has limitations in terms of interpreting disease relevance, I think the direct observation that the MSP domain is somehow able to sense or respond to a signal about the cellular metabolic state is a crucial and unexpected finding that will have long lasting implications for our field. There are a few minor technical issues (detailed below), and I feel the claims about understanding disease mechanisms in the text could be dialed back a bit, but in general I am very enthusiastic about this work. I look forward to seeing it press, as it will stimulate lots of more easily interpretable and controllable experiments in human cells.

Minor Points:

1. It would be nice to see some type of control for the amount of VAPB or P56S VAPB present under these conditions. Since we know elevated P56S VAPB expression causes aggregation, it would be good to know if any of the aggregation phenotypes throughout this paper represented changes in expression of the molecule itself. Since the molecules are GFP-tagged, this should be a relatively trivial control to add, and I feel it is of particular importance in the experiments in Fig. 1 where there is a lot of time for something like expression level to be adjusted.
2. I think the authors use the word "physiological" to describe expression in this system a little too freely. Physiological for what? For yeast? Or for what kind of human cell? I understand they are trying to avoid criticisms about relevance, but it strays into the realm of being misleading for less experienced readers. This system models at a level (and with a C-terminal fusion from SCS2 that is sort of downplayed in the main text) that works for yeast-and that is part of what makes it so interesting. There is no need to overplay the disease relevance.
3. One Line 209, the authors reference some of their previous work as showing that P56S marks cisternae. While this is technically accurate, it feels a bit misleading-If I recall correctly, the majority of the P56S-associated structures in Fasana 2010 are more appropriately described as OSERs-not the neat cisternae shown here. Do the authors think this is because of the limited amount of P56S expressed in this system? It may be worth stating something about this specifically.
4. The two color system introduced in Figure 4 is amazing. It would be wonderful to see some quantification of the extent to which the two forms interplay, since the authors already have the data.
5. Figure 2D is missing a size ladder.
6. Figure 2D What is the specificity of bands? It's not clear what we are looking at.
7. Fig 5D, why no error bars?
8. Line 325, the incorrect figure is referenced.

We sincerely thank the reviewers for their thoughtful feedback, insightful comments, and valuable suggestions. We have carefully addressed all of their comments, as outlined below.

Reviewer #1

Comments to the authors:

The manuscript entitled "A role of ER-mitochondrial contact sites in amyotrophic lateral sclerosis 8 (ALS8) pathogenesis" by C. Wilson et al., describes the observations that can be made when human VAPB, the protein mutated in ALS8 is expressed in the model organism *S.cerevisiae*. To achieve "physiological" levels of VAPB the gene encoding the yeast paralogue *SCS2* was replaced by WT and mutant VAPB by homologous recombination. The main message of the study is that in yeast the mutated VAPB favors inclusion/aggregate formation at the ER that is evoked by a mitochondrial aberration. The authors claim that the mitochondrial changes are relayed to the ER by mitochondrial-ER contact sites. The observations made are quite interesting and the study is well executed. However, I have some concerns that should be addressed.

1) "Physiological expression levels of VAPB in yeast". In my opinion it is rather difficult to determine such an expression level for a paralogue from another organism. Not only expression by a certain promoter determines protein levels but also e.g. protein stability or posttranslational modification.

The authors should construct a yeast strain that contains *Scs2p* with a GFP-module similarly inserted next to the TMD as done for VAPB. Using these cells at least similar protein-levels of *Scs2p* and VAPB can be shown.

R. We agree with the referee that the expression of the human VAPB will always be affected by factors such as protein stability or posttranslational modifications. Analysis of the expression levels of the yeast (*Scs2*) and human (VAPB) GFP-tagged forms would permit us to compare the levels of the endogenous and human forms. However, the aim of our work was not to obtain exactly the same level of expression of the transgene compared with the endogenous one, but rather to use a cell system, in which the human protein would not be severely overexpressed, as occurs in commonly used expression systems. In light of these considerations, we decided to remove the expression "ensuring expression at or near physiological levels" and replaced it with "... under the control of the promoter and terminator of the yeast homolog *SCS2*, at the *SCS2* locus, in order to prevent overexpression of the transfected proteins." (page 3).

The functionality assay (FigS1A) is not really convincing since although WT-VAPB shows some rescue of growth overall (approx. 50-60%) in comparison to *scs2d* cells, the single colonies are

rather small in comparison to BY4742 WT, reminiscent of petite colonies formed by yeast cells that have respiratory/mitochondrial defects. To show loss of function of the P56S mutant the authors should show by Western blot that the cells of FigS1A have comparable VAPB (WT or P56S) expression. By this a lower P56S expression/stability in the cells responsible for the lack of rescue, can be excluded.

R. The levels of WT or P56S VAPB under conditions of inositol depletion cannot be compared, since the P56S-VAPB expressing cells do not grow under this condition and as a consequence it is impossible to prepare cell extracts for analysis. However, WT and P56S VAPB proteins showed comparable levels of expression in all experiments, where inositol was present. For instance the Western blot in Fig. 2A shows that WT and P56S VAPB (both in their untagged and GFP-tagged forms) have approximately the same expression levels when grown in SD+complete medium (which is the same medium as the plates shown in Fig. S1A, left panel). Although we cannot exclude that instability of P56S-VAPB contributes to its incapacity to rescue cells from inositol auxotrophy, we think it highly unlikely that some reduction in its levels could account for its complete lack of functionality in the assay. In any case, what the figure wants to show is simply that the mutation, like in mammals, has severe functional consequences in yeast. A similar finding was reported by Suzuki et al. (2009), with the use of a different expression system. While there is no difference in growth rate between cells expressing WT-VAPB or P56S-VAPB during exponential growth in rich medium, WT-VAPB can complement (though partially, as mentioned in the text) the loss of Scs2 in terms of inositol auxotrophy, while P56S has no activity. Our data are in accordance with the complementation data of Stump et al. (2023), who also found that WT-VAPB only partially complements the loss of Scs2.

In response to the referee's comment that the single colonies under conditions of rescue from inositol auxotrophy are small in comparison to the WT cells and resemble petite colonies, we would like to point out that cells expressing WT-VAPB cannot be formally defined as petite since they can grow on media containing only non-fermentable carbon sources (such as glycerol). Since rescue from inositol auxotrophy is partial, the smaller colony size can be attributed to the slower growth rate of the cells under this condition.

How highly conserved is VAPB to SCS2 and SCS22?! Would it be possible to introduce the P56S mutation in Scs2 or Scs22 and mimic/observe the inclusion/aggregation with the yeast protein?!

R. This is an interesting question that has been addressed by Nakamichi et al. (PMID: 21144830) who examined the sequence alignment of VAPs around the VAP consensus sequence (VCS, which includes P56) and found a difference in the proline distribution among VAPs. Most VAPs have three prolines around the VCS, whereas VAPB has only two prolines in this region. The difference is notable because the Pro-56 mutation of VAPB leaves only one proline within the VCS, whereas Scs2p (and VAPA) retain two prolines even if the proline equivalent to the Pro-56

(Pro-51 and Pro-56, respectively) is substituted with a different amino acid. Importantly substituting p51 in *scs2* (equivalent to p56 in VAPB) or p56 in VAPA does not change the behavior of the protein. The authors concluded that the appropriate distribution of three conserved prolines, not the existence of a particular proline, confers Scs2p (and VAPA) resistance to the Pro-56 mutation.

2) Protein steady state levels should be determined by Western Blot with antibodies against VAPB or GFP (if the P56S mutation alters ab affinity) for all experiments where the WT VAPB is compared to the mutant for forming the aggregates/inclusions.

R. As mentioned above, we have systematically determined the protein levels in the experiments shown in the manuscript (and we now specify it in the manuscript under the method section). We have also ascertained, by comparison with an anti-GFP Ab, that the anti-VAPB Ab recognizes WT and P56S with the same affinity. As stated above the levels of WT and P56S VAPB proteins, both at steady state conditions and after NaN_3 treatment, are comparable (as shown in Fig. 2A and below in Fig. 1). They remain comparable also when taking into consideration internal loading controls (i.e. Ponceau staining and Pgk1 (Fig. 1)).

Figure 1 Cells expressing WT-VAPB or P56S-VAPB, untreated or treated with NaN_3 , were processed using the Triton-X 100 extraction protocol. The membranes were stained with Ponceau S and then probed with anti-Pgk1 and anti-VAPB antibodies. TL = total lysate; S21 = 21,000g supernatant; P21 = 21,000g pellet (microsomal fraction); S21T = 21,000g supernatant after Triton X-100 extraction of the P21 pellet; P21T = 21,000g pellet after Triton X-100 extraction of the P21 pellet.

3) For all conditions where the inclusions were observed it should be confirmed, that one is dealing with the same type of protein inclusions, e.g. the TX-100 resistance assay.

R. The Triton-X extraction protocol was performed several times and under different treatments (see examples in Fig. I and Fig. II A-D). However, since the results indicated the same response and the microscopy indicated the same ER-associated structures, we concentrated on the NaN_3 treatment for the full analysis, as it provided the most robust and controllable response in terms of inclusion formation. We provide below multiple examples of our analysis of protein levels and triton extractability of VAPB in microsomal fractions.

Figure II Triton-X 100 extraction protocol of haploids (A, B, D) and diploids (C). Cells were treated with NaN_3 in A-C. Different anti-VAPB antibodies were used in A (in-house) and B (commercial). In C, cells expressed either tagged and untagged P56S-VAPB (left) or tagged and untagged WT-VAPB (right). D. Cells expressing P56S-VAPB were glucose-depleted or galactose-depleted with the addition of antimycin A, and then processed using the Triton-X 100 extraction protocol. TL = total lysate; S21 = 21,000g supernatant; P21 = 21,000g pellet (microsomal fraction); S21T = 21,000g supernatant after Triton X-100 extraction of the P21 pellet; P21T = 21,000g pellet after Triton X-100 extraction of the P21 pellet.

4) Figure 2B: The same experiment should be performed with WT VAPB to see what the ER looks like without the inclusion formation.

R. The fluorescence imaging of the ER in WT-VAPB expressing cells showed a regular appearance with no aggregates, so we lacked a “reference fluorescent object” to be analyzed by CLEM. However, given the regular appearance of fluorescence signal of WT-VAPB, we anticipate that the ER of WT-VAPB expressing cells looks like generic ER by electron-tomography

(PMID: 21502358). In support of this, in a previous publication, we have analyzed ER marked by GFP-Scs2 by CLEM and found generic, cortical ER (PMID: 31743663).

5) Figure 2C: Why is a higher molecular weight adduct (via disulfide bridges) forming under reducing conditions of the cytosol (the N-terminus of VAPB faces the cytosol). Could this be during the lysis of the cells?

R. The legend to Figs. 2C and D of the original manuscript was not clear, and we have fixed this in the revised version. Fig. 2C reports on cell-free interactions between purified recombinant proteins. Samples were all reduced before loading on the gel, so no high molecular adducts could be observed in these gels. The 31 kDa band corresponds to the monomeric recombinant fusion protein. Mutant, but not WT MSP was captured by bead-bound His-SUMO-MSP. If reducing agent (DTT) was present during the pull-down incubation, untagged MSP (either WT or P56S) was not recovered with the beads, suggesting that disulfide bonding is involved. A different analysis is presented in Fig. 2D, where the state of the MSP domains recovered from *E. coli* lysates is analyzed either without reduction or after addition of a reducing agent to the samples before loading on the gels. The panel shows that the mutant MSP is recovered as high molecular weight DTT-sensitive multimers. Altogether, the results of Figs. 2C and D demonstrate that, differently from WT MSP, P56S-MSP has a propensity to form higher order disulfide-bonded species.

6) Are the punctate structures observed in case of the P56S mutant ER-bound or protein aggregates in the cytosol. To this end the authors should show by cell fractionation experiments that WT and mutant VAPB can be found in the ER (microsome) fraction.

R. Different lines of evidence indicate that the punctate structures containing P56S VAPB are in the ER. First, as the reviewer suggests, we performed cell fractionation and analyzed the microsome fraction (indicated as p21: pellet obtained after 21000g centrifugation) for its Tritonx100-extractability (Fig.2A). We realize that this was not clearly described in the legend (but only in methods) and now we made it more explicit also in the legend to Fig. 2A. Figure II above provides further multiple examples of the analysis of the microsomal fraction (p21). Second, aggregates do not form when the TMD is missing from the protein (data not shown) so association with the ER is required. Third, it is clearly visible from the fluorescent images in the manuscript and the additional ones shown below (Fig. III), that the aggregates are associated with the perinuclear and cortical ER.

Figure III Fluorescent images of glucose-depleted cells (left) and NaN_3 -treated cells (right) show that aggregates are associated with the perinuclear and cortical ER.

7) Figure 3 A: The combination with a mutant variant should be shown also in log phase
 Figure 3B: The WT combination (Fig3A) should also be analyzed and shown under glucose red.
 Figure 3C: The WT combination (Fig3A) should also be analyzed and shown under N3- treatment.

R. Following the referee’s suggestion, we have modified Fig. 3 and Fig. S2. Fig. 3A now includes images of the combination WT-VAPB-GFP/P56S-VAPB-RUBY in log phase, and Fig. 3B the images of WT combination in response to glucose deprivation, while Fig. S2 shows the WT combination in response to glucose deprivation and NaN_3 treatment. As stated in the manuscript, the WT protein never forms inclusions under any of the conditions tested.

8) Figure 6: Since deletion of different mitochondrial morphology genes have different impact on the growth phenotype of yeast cells, which in turn might impact protein expression/stability the expression levels of VAPB-P56S should be compared for all of the mutants.

Also especially in those deletion mutants that are clearly different from WT (ERMES/Num1) the behavior of WT-VAPB should be analyzed.

R. As the referee rightly pointed out, certain mutants impact on the growth phenotype of yeast cells. However, the inclusion formation does not correlate with the expression levels of the mutant protein, as shown below in Fig. IV reporting the levels of P56S in mutant strains that show less (mdm34) or more (num1) inclusions while expressing comparable levels of P56S-VAPB protein.

Figure IV Ponceau staining of cell extracts from the indicated strains and WB of VAPB. Numbers on the top indicate the level of P56S-VAPB in each strain, normalized for the total proteins of the ponceau stain and reported as ratio respect to P56S-VAPB in the WT strain.

The behavior of WT-VAPB has been analyzed in the *num1* mutant strain, both in log phase and under NaN_3 treatment and no inclusions have been observed (Fig. V).

Figure V WT-VAPB-GFP in the *num1* strain under log phase and NaN_3 treatment.

9) Figure S5: ERMES subunits were reported to show multiple foci on mitochondria in fluorescence-microscopy. Therefore, the uniform cytosolic distribution of Mdm34-GFP is rather unusual. The authors should provide (grayscale) pictures, where the usual ERMES distribution can be observed.

R. To address the reviewer comments we have modified Figure S5 and introduced grayscale images that better show the ERMES distribution.

Reviewer #2

The P56S mutation in the MSP domain of VAPB causes a familial form of ALS (ALS8). The mechanism(s) through which this mutation results in disease remains elusive in spite of intense investigation. This condition has dominant transmission, and there has been evidence for both toxic gain of function or haploinsufficiency. It has been reported that VP56S VAPB can form aggregates in cells, most likely due to misfolding, but formation of these aggregates varies with experimental conditions. Moreover, the link of aggregates to disease remains unclear. In this study the authors developed a yeast model to study conditions that lead to P56SVAPB aggregation. They find that mitochondrial dysfunction facilitates aggregate formation and that disruption of ER-mitochondria tethers reduce aggregate formation. Based on these findings they propose that that some signal transmitted to the ER from damaged mitochondria may facilitate aggregate formation. They also report that co-expression of the WT protein also reduced aggregate formation, suggesting a protective effect.

These findings advance knowledge about P56SVAPB pathology and will be of interest in the ALS field. However, many open questions remain and I find the key conclusion "the role of ER-mitochondria contacts in pathology" premature based on the data shown. I suggest to present this idea as a hypothesis, not as a conclusion and to remove it from the title.

R. We have modified the title as suggested by the referee: "A role for mitochondrial-ER crosstalk in amyotrophic lateral sclerosis 8 pathogenesis"

Major comments

The authors quote in the introduction the previous study by Stump et al (2023) which capitalized on an approach similar to the one used here to study the impact of the Als8 mutation in yeast. It would be helpful if the authors discussed in more detail the relations of their findings to those of Stump et al.

R. The study of Stump et al. used doubly-deleted (*scs2/scs22*) cells and introduced the VAPB genes at the *Leu2* locus. We used only *scs2* deletion, by replacing *Scs2* with VAPB at the *Scs2* locus to try to mimic the disease situation as closely as possible. Differently to Stump et al., we were not aiming to assess VAPB substitution of *Scs2* function *per se*, rather, we wanted to use the yeast model to gain insights into the property of the P56S-VAPB protein to form inclusions. As a consequence, the types of assays we used were very different to those of Stump et al. Following the reviewer's suggestion, we have now discussed in more detail in the manuscript the aims and results of the two studies at page 10.

Evidence that reduction of ER-mitochondria contacts has an attenuating effect on pathology is preliminary. The authors base this conclusion primarily on data involving disruption of the ERMES complex in yeast and of the VAPB - PTPIP51 interaction in mammalian cells. But ERMES (which does not exist in mammals) and the contacts between ER and mitochondria mediated by VAPB and PTPIP51 have very different functions. ERMES does not exist in mammalian cells, and PTPIP51 is unlikely to fulfill ERMES's function based on its very different structure and properties. This should at least be mentioned. Moreover, PTPIP51 directly interacts with VAP. Thus, there may be compounding effects. The strong emphasis on the role of ER-mitochondria contacts should be downplayed, unless this topic is further developed.

R. Following the reviewer suggestion, we have reduced the emphasis on ER-mito contacts, by modifying the title, eliminating a paragraph on the parallelism between ERMES and PTPIP51

(discussion) while introducing a critical appraisal of the difference between ERMES and PTPIP51 (page 10).

Experiments were carried out in yeast cells where the *SCS2* gene was replaced by human VAPB. Both in humans and in yeast, there are two VAP genes: VAPA and VAPB in mammals and *SCS2* and *SCS22* in yeast. In the experiments focused on dimerization, can the author comment on the potential confounding effect of the other isoform?

R. The experiments shown in Fig. 2 C-E were performed with the isolated recombinant proteins (WT and P56S-VAPB) expressed in *E. coli* with the sole aim of showing the different propensity of the two forms of VAPB to interact “homotypically” as no VAPA was present under these conditions. However, we did test the intriguing possibility suggested by the reviewer (see below) that VAPA may play a role in controlling the P56S-VAPB oligomerization/inclusion formation, and we found that this is the case (Fig. 4D of the revised manuscript).

The MSP domain of VAPB can dimerize (PMID: 16004875), as also reported in this study. One would expect that physiological dimerization occurs "in cis", i.e. dimerization of two VAP proteins on the same membrane. The dimerization proposed here to mediate ER aggregation is expected to occur "in trans", on two closely apposed ER membranes. As WT VAP does not induce these appositions, the dimerization generated by P56SVAPB is likely to be mediated by a different type of dimerization. The authors should comment on this.

R. We thank the reviewer for this comment. Indeed, the oligomerization properties of WT and P56S-VAPB proteins were compared by Kim et al. (PMID: 20207736). They found that the oligomerization of the WT-VAPB is independent of its MSP domain but requires the coiled-coil domain, while the P56S mutation, possibly inducing conformational changes within the MSP domain, facilitates its propensity to aggregate. Thus, we hypothesize that the dimerization of the WT-VAPB involving mainly the coiled-coil domain close to the TM domain mediates the “in cis” interaction, while the oligomerization of P56S-VAPB involving the cytosolic MSP domain mediates the “in trans” interaction. We have introduced these considerations in the manuscript at page 12.

WT-VAPB is reported here to be protective on the formation of aggregates induced by the P56SVAPB mutations. It would be interesting to know if VAPA is also protective.

R. We have addressed this interesting point raised by the reviewer by silencing VAPA in B1 NSC34 cells, in three independent experiments. As now shown in Fig. 4D, the number of cells

forming VAPB-P56S aggregates increases when VAPA is downregulated, suggesting a protective role of VAPA on the formation of aggregates induced by the P56S-VAPB mutation.

Reviewer #3

Wilson et al. introduce an interesting series of experiments based on yeast lines in which they have replaced the major cytoplasmic domains of endogenous VAPB ortholog SCS2 with GFP-tagged human VAPB with or without the ALS-causative mutation P56S. They make a number of interesting observations, including that P56S inclusions are only induced under specific growth conditions, that this is dependent on mitochondrial function, and that it is most affected by sites of contact with the endoplasmic reticulum, rather than as one might expect on ATP levels or other obvious metabolic signals.

Although the use of Yeast as a model system certainly has limitations in terms of interpreting disease relevance, I think the direct observation that the MSP domain is somehow able to sense or respond to a signal about the cellular metabolic state is a crucial and unexpected finding that will have long lasting implications for our field. There are a few minor technical issues (detailed below), and I feel the claims about understanding disease mechanisms in the text could be dialed back a bit, but in general I am very enthusiastic about this work. I look forward to seeing it press, as it will stimulate lots of more easily interpretable and controllable experiments in human cells.

Minor Points:

1. It would be nice to see some type of control for the amount of VAPB or P56S VAPB present under these conditions. Since we know elevated P56S VAPB expression causes aggregation, it would be good to know if any of the aggregation phenotypes throughout this paper represented changes in expression of the molecule itself. Since the molecules are GFP-tagged, this should be a relatively trivial control to add, and I feel it is of particular importance in the experiments in Fig. 1 where there is a lot of time for something like expression level to be adjusted.

R. Indeed, we systematically checked the levels of the expressed proteins by WB (and found that WT and P56S-VAPB are expressed at comparable levels (and we now specify it in the manuscript under the methods section). In particular, the protein levels referring to experiments in Fig. 1, both under steady state conditions and after NaN₃ treatment, are comparable (as shown in Fig. 2A). They remain comparable also when taking into consideration to internal loading controls (i.e. Ponceau staining and Pgk1).

We provide below (Figs. I and II) multiple examples of our analysis of protein levels and Triton extractability of the VAPB proteins in microsomal fractions that show that WT and P56S-VAPB

proteins are expressed in comparable amounts under different experimental conditions. Therefore, we would exclude overexpression of the protein to be a contributing factor to aggregation.

Figure I Cells expressing WT-VAPB or P56S-VAPB, untreated or treated with NaN_3 , were processed using the Triton-X 100 extraction protocol. The membranes were stained with Ponceau S and then probed with anti-Pgk1 and anti-VAPB antibodies. TL = total lysate; S21 = 21,000g supernatant; P21 = 21,000g pellet (microsomal fraction); S21T = 21,000g supernatant after Triton X-100 extraction of the P21 pellet; P21T = 21,000g pellet after Triton X-100 extraction of the P21 pellet.

Figure II Triton-X 100 extraction protocol of haploids (A,B,D) and diploids (C). Cells were treated with NaN_3 in A-C. Different anti-VAPB antibodies were used in A (in-house) and B (commercial). In C, cells expressed either tagged and untagged P56S-VAPB (left) or tagged and untagged WT-VAPB (right). D. Cells expressing P56S-VAPB

were glucose-depleted or galactose-depleted with the addition of antimycin A, and then processed using the Triton-X 100 extraction protocol. TL = total lysate; S21 = 21,000g supernatant; P21 = 21,000g pellet (microsomal fraction); S21T = 21,000g supernatant after Triton X-100 extraction of the P21 pellet; P21T = 21,000g pellet after Triton X-100 extraction of the P21 pellet.

2. I think the authors use the word "physiological" to describe expression in this system a little too freely. Physiological for what? For yeast? Or for what kind of human cell? I understand they are trying to avoid criticisms about relevance, but it strays into the realm of being misleading for less experienced readers. This system models at a level (and with a C-terminal fusion from SCS2 that is sort of downplayed in the main text) that works for yeast-and that is part of what makes it so interesting. There is no need to overlay the disease relevance.

R. We have removed “ensuring expression at or near physiological levels” and replaced it with “... under the control of the promoter and terminator of the yeast homolog SCS2, at the SCS2 locus, in order to prevent overexpression of the transfected proteins.”

3. One Line 209, the authors reference some of their previous work as showing that P56S marks cisternae. While this is technically accurate, it feels a bit misleading-If I recall correctly, the majority of the P56S-associated structures in Fasana 2010 are more appropriately described as OSERs-not the neat cisternae shown here. Do the authors think this is because of the limited amount of P56S expressed in this system? It may be worth stating something about this specifically.

R. As the reviewer points out we are comparing different experimental models: mammals vs yeast and high vs lower levels of expression on P56S-VAPB. We have introduced this consideration in the manuscript (page 5).

4. The two color system introduced in Figure 4 is amazing. It would be wonderful to see some quantification of the extent to which the two forms interplay, since the authors already have the data.

R. We have modified Fig. 3, now showing the double-colored diploid cells expressing all the different combinations of WT and/or P56S and we have performed a quantitative analysis of the cells with inclusions upon glucose depletion (Fig. 3C).

5. Figure 2D is missing a size ladder.

R. We have introduced the MW standards.

6. Figure 2D What is the specificity of bands? It's not clear what we are looking at.

R. We have explained more extensively the experiment and the resulting protein bands in the text and in the legend. We have modified Fig. 2D introducing a Western blot of WT and P56S-SUMO-His proteins purified from bacteria, immunodetected with an anti-histidine antibody, in order to distinguish specifically VAPB proteins from bacterial contaminants. The panels clearly show that WT MSP is mostly in its monomeric and dimeric forms, while the P56S MSP runs in multiple oligomeric forms which are mostly resolved upon DTT treatment.

7. Fig 5D, why no error bars?

R. We have added the error bars.

8. Line 325, the incorrect figure is referenced.

R. We have replaced Fig. 4E with the correct figure reference (Fig. S4E).

December 24, 2024

RE: Life Science Alliance Manuscript #LSA-2024-02907-TR

Prof. Maria Antonietta De Matteis
Telethon Institute Of Genetics And Medicine
Via Campi Flegrei 34
Pozzuoli, Naples 80078
Italy

Dear Dr. De Matteis,

Thank you for submitting your revised manuscript entitled "A role for mitochondria-ER crosstalk in amyotrophic lateral sclerosis 8 pathogenesis". We would be happy to publish your paper in Life Science Alliance pending final revisions necessary to meet our formatting guidelines.

- please be sure that the authorship listing and order is correct
- please use the [10 author names, et al.] format in your references (i.e. limit the author names to the first 10)

Figure Check:

- please add sizes next to all blots

A. FINAL FILES:

B. MANUSCRIPT ORGANIZATION AND FORMATTING:

**Submission of a paper that does not conform to Life Science Alliance guidelines will delay the acceptance of your

manuscript.**

The license to publish form must be signed before your manuscript can be sent to production. A link to the electronic license to publish form will be available to the corresponding author only. Please take a moment to check your funder requirements.

Sincerely,

Reviewer #1 (Comments to the Authors (Required)):

My concerns raised in the first review process were addressed sufficiently in the revised version of the manuscript.

January 8, 2025

RE: Life Science Alliance Manuscript #LSA-2024-02907-TRR

Prof. Maria Antonietta De Matteis
Telethon Institute Of Genetics And Medicine
Via Campi Flegrei 34
Pozzuoli, Naples 80078
Italy

Dear Dr. De Matteis,

Thank you for submitting your Research Article entitled "A role for mitochondria-ER crosstalk in amyotrophic lateral sclerosis 8 pathogenesis". It is a pleasure to let you know that your manuscript is now accepted for publication in Life Science Alliance. Congratulations on this interesting work.

DISTRIBUTION OF MATERIALS:

Again, congratulations on a very nice paper. I hope you found the review process to be constructive and are pleased with how the manuscript was handled editorially. We look forward to future exciting submissions from your lab.

Sincerely,
